# Superior colliculus saccade motor bursts do not dictate movement kinematics

Tong Zhang [1,2], Tatiana Malevich[1,2], Matthias P. Baumann[1,2] & Ziad M. Hafed [1,2✉]

The primate superior colliculus (SC) contains a topographic map of space, such that the anatomical location of active neurons defines a desired eye movement vector. Complementing such a spatial code, SC neurons also exhibit saccade-related bursts that are tightly synchronized with movement onset. Current models suggest that such bursts constitute a rate code dictating movement kinematics. Here, using two complementary approaches, we demonstrate a dissociation between the SC rate code and saccade kinematics. First, we show that SC burst strength systematically varies depending on whether saccades of the same amplitude are directed towards the upper or lower visual fields, but the movements themselves have similar kinematics. Second, we show that for the same saccade vector, when saccades are significantly slowed down by the absence of a visible saccade target, SC saccade-related burst strengths can be elevated rather than diminished. Thus, SC saccade-related motor bursts do not necessarily dictate movement kinematics.

[1] Werner Reichardt Centre for Integrative Neuroscience, Tübingen University, 72076 Tübingen, Germany. [2] Hertie Institute for Clinical Brain Research, Tübingen University, 72076 Tübingen, Germany. ✉email: ziad.m.hafed@cin.uni-tuebingen.de

The superior colliculus (SC) plays an important role in saccade generation, as evidenced by the ease with which low-current electrical microstimulation of SC neurons evokes saccades[1,2]. Anatomically, SC neurons are organized to form a spatial code of eye movement displacement vectors[1,3,4], such that the location of an active neuron in the SC defines the amplitude and direction of a desired saccade. Robustness and accuracy of saccade vector representation are ensured through population coding[4–6], with the aggregate activity of a large number of simultaneously active neurons defining a given movement's metrics.

The SC spatial code necessarily entails a temporal synchrony of SC activity at the time of saccades. Indeed, saccade-related neurons show a highly characteristic temporal evolution of spiking[7–11], dominated by a burst tightly locked to movement onset. Interestingly, the strength of such a burst can vary, suggesting that SC neurons may encode additional properties beyond the saccadic displacement vector represented by the spatial code. For example, blink-perturbed saccades can have weaker, but prolonged, bursts[12,13]. Moreover, burst evolution during a saccade may be related to the remaining motor error of an ongoing eye movement[14,15] (i.e., how much more the eye needs to keep moving), or it may be related to the speed profile of the ensuing saccade[16]. Additionally, saccade-related burst strength can be modulated by audio-visual sensory combinations[17]. Thus, there is an SC rate code for saccades, the role of which is less well understood than that of the spatial code.

The most recent SC models posit an important role for the rate code in dictating saccade kinematics[13,16,18]. In these models, the locus of an active neuron (i.e., the spatial code) defines how each individual spike in a motor burst moves the eye along the amplitude dimension; parameters like eye speed or time to movement end would reflect the strength of (i.e., number of spikes in) the motor burst (i.e., the rate code). While appealing in their combination of both spatial and rate codes for movement specification, these models suggest a very tight relationship between saccade-related burst strength and movement kinematics. However, this may not necessarily always be the case. For example, we recently explored a situation in which saccade kinematics were altered by a simultaneity condition between a motor burst somewhere on the SC map and an irrelevant visual burst somewhere else[19,20]. When we recorded at both the motor and visual burst locations[21], we found a lawful relationship between the saccade changes and the number of additional spikes injected by the visual burst (consistent with the spatial code); however, critically, the simultaneous motor burst was minimally affected[21]. Thus, the rate code of the original movement commands was essentially unaltered even though the movements themselves were. This, along with other evidence[22], motivates investigating whether saccade kinematics are indeed dictated by the SC rate code or not.

We approached this question using two complementary approaches. In the first, we exploited a large asymmetry in how the SC represents the upper versus lower visual fields in its visual sensitivity[23]. If such an asymmetry still holds, but now for saccade-related motor burst strength, then there should be (at least according to current models of the rate code) systematic differences in the (amplitude-matched) saccades' kinematics. We confirmed a neural asymmetry in SC motor burst strengths, but found no concomitant kinematic differences between amplitude-matched saccades towards the upper and lower visual fields. In the second approach, we instead used vector-matched saccades, but of clearly different kinematics. Specifically, we exploited the fact that saccades to a blank can have significantly slower speeds than saccades to a clear, punctate visual target[24–30]. We, therefore, compared SC neuron motor bursts in these two conditions,

sometimes recording multiple neurons simultaneously in the two behavioral contexts. Surprisingly, we found no correlation between SC motor burst modifications and the kinematic alterations of the saccades. More importantly, approximately one quarter of the neurons actually increased, rather than decreased, their motor burst strengths for the slower saccades. Our observations highlight the need to explore other potential functional roles for the saccade-related SC rate code.

## Results

We first identified a dissociation between SC motor burst strengths and their associated eye movements' kinematics. Specifically, we explored how SC motor burst strength might differ as a function of visual field location. When we recently described an asymmetry in how the SC represents the upper and lower visual fields[23], we found that SC visual response properties were different across the fields. We also briefly mentioned that the strength of saccade-related motor bursts may also be asymmetric[23]. Here, we investigated the robustness of this saccade-related neural asymmetry in more detail, and we then asked whether it predicted an asymmetry in amplitude-matched saccade kinematics between movements towards the upper and lower visual fields. That is, we were motivated by a common assumption in existing models[13,16,18] that the relationship between SC motor burst strengths and movement kinematics is directionally symmetric and only depends on movement amplitudes. If so, then amplitude-matched movements of different directions, which have different SC motor burst strengths, should also have different kinematics. In a second set of experiments, we then used the complementary approach: we compared vector-matched saccades (where both the amplitude and direction were the same and towards the response field hotspot), and we asked whether alterations in these saccades' kinematics under different behavioral contexts were systematically related to alterations in the SC motor burst strengths.

In what follows, we first describe the amplitude-matched upper and lower visual field saccade results, and we then turn to the experiments with the vector-matched saccades having different kinematics.

**Difference in superior colliculus motor burst strengths for saccades towards the upper versus lower visual fields.** Visual sensitivity is significantly stronger in SC neurons representing the upper visual field[23]. That is, if a neuron's visual response field has a preferred (hotspot) location above the retinotopic horizontal meridian and we present a stimulus at this location, then the neuron's response is stronger than that of a neuron with a lower visual field hotspot location (and a stimulus presented at its preferred location). Curiously, in our earlier study[23], we noticed that saccade-related motor bursts showed the opposite asymmetry: saccade-related motor bursts (for preferred hotspot locations) were stronger for neurons representing saccades towards the lower visual field than for neurons representing saccades towards the upper visual field. However, in that study[23], we did not control for the depths of the recorded neurons from the SC surface when we analyzed the neurons' motor bursts. Since the strength of SC motor bursts can vary with depth from the SC surface[6] (for a given range of tested eccentricities), here, we wanted to first confirm whether the asymmetry alluded to above[23] was still present when carefully controlling for neuron depth ("Methods"). If this was the case, we could then ask whether saccade kinematics were systematically different or not.

We re-analyzed the neural database of ref. [23] (monkeys P and N) by first matching the depths of neurons from the SC surface between the upper and lower visual fields ("Methods"). For each

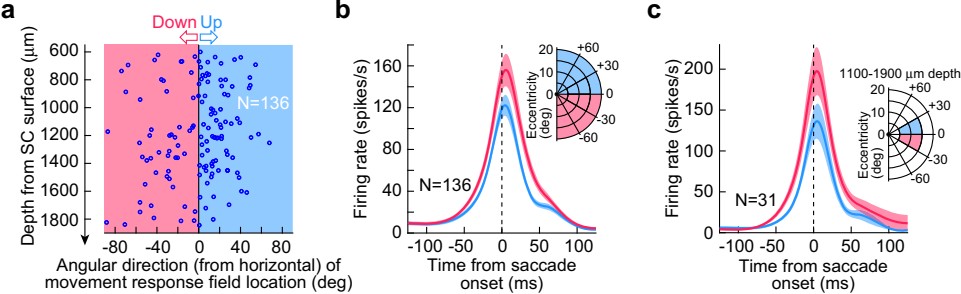

**Fig. 1 Superior colliculus (SC) saccade-related motor bursts are stronger for downward saccades than for upward saccades, even when controlling for depth from the SC surface. a** We picked neurons matched for depth from the SC surface (between 600 and 1850 μm) but having movement field hotspot locations in either the upper (light blue) or lower (light red) visual fields (i.e., positive or negative directions from the horizontal meridian, respectively). Note that the upper visual field neurons appear compressed along the direction dimension (i.e., in visual coordinates), likely due to upper visual field neural tissue magnification[23]. Such magnification is similar, in principle, to foveal magnification of SC neural tissue[33]. **b** For the neurons in (**a**), we plotted peri-saccadic firing rates for saccades towards each neuron's preferred movement field location[23]. Saccade-related bursts were stronger for lower visual field than upper visual field neurons. **c** To better constrain errors in depth estimates from the SC surface (due to surface curvature), we further restricted the choice of neurons to those primarily near the horizontal meridian and at an eccentricity range associated with quasi-constant tissue curvature between upper and lower visual field locations[33]; the ranges of amplitudes, directions, and depths are shown in the inset. The motor bursts of the resulting 31 neurons were still stronger for lower visual field than upper visual field neurons (left panel). Error bars in all panels denote s.e.m.

extra-foveal neuron in this database (here referred to as database 1), we identified whether the neuron was saccade-related or not[23]. We then classified the saccade-related neurons as having a movement-related preferred response field location (or hotspot location) in the upper or lower visual field. Preference was defined as the location for which saccades were associated with the highest firing rates, similarly to how visual preference was defined as the location for which visual stimuli evoked the strongest visual bursts[23]. Finally, we picked a range of neural depths from the SC surface that was overlapping between the upper and lower visual field neurons. This final step was the critical step for the present analysis, and it resulted in us having a total of 136 SC neurons with depths from the SC surface between 600 and 1850 μm ("Methods"). The distribution of these depths is represented in Fig. 1a, where each neuron's depth is plotted against a measure of whether the neurons' preferred movement-related response field location was above or below the horizontal meridian (the x-axis shows the angular direction of the preferred location from the horizontal meridian; positive means above the meridian, and negative means below). As can be seen, we found saccade-related SC activity at a range of depths from the SC surface that was consistent with prior observations[6–10,31,32]. Critically, the neural depths were overlapping between the upper and lower visual field neurons (Fig. 1a; $p = 0.1257$, t-test, t-statistic: −1.5408, df: 134); note (as an aside) that upper visual field directions were compressed relative to lower visual field directions, which is consistent with the idea of upper visual field neural tissue magnification in the SC[23]. Therefore, we were now in a position to check whether an asymmetry of saccade-related burst strengths alluded to earlier[23] still held after controlling for neuron depth.

Having established that we now had a neural database with matched depths from the SC surface, we proceeded to comparing motor burst strengths between the upper and lower visual field neurons. Here, we exploited the similar ranges of eccentricities covered by our upper and lower visual field neurons, as well as the expectation that SC motor burst strengths should be fairly similar, within either the upper or lower visual fields, for these tested ranges. We plotted the peri-saccadic firing rates of the neurons of Fig. 1a, from a delayed, visually-guided saccade task ("Methods"). We employed such a delayed saccade paradigm to allow analyzing motor bursts in isolation, without the recently occurring visual bursts associated with target onset, which would have come too close to saccade onset in an immediate, visually-guided saccade

version of the task ("Methods"). We picked, for each neuron, the preferred saccades of the neuron and plotted its firing rate for these movements, as we did previously[23]. We then averaged across all neurons (Fig. 1b). There was indeed an asymmetry in SC motor burst strengths, such that neurons representing the lower visual field had significantly stronger motor bursts than neurons representing the upper visual field (Fig. 1b), despite the neurons covering a similar range of preferred eccentricities in the upper and lower visual field groups. To statistically assess the difference in burst strengths after matching for neural depths from the SC surface, we measured the average firing rate in the final 50 ms before saccade onset for each neuron's preferred saccades[23] ("Methods"). We then compared the population of measurements for the upper and lower visual field neurons of Fig. 1a using a t-test. Across neurons, average firing rate for the upper visual field neurons was 99 spikes/s, and it was 121 spikes/s for the lower visual field neurons. This difference was statistically significant ($p = 0.039$, t-test, t-statistic: −2.0844, df: 134). Therefore, even after controlling for the depths of neurons from the SC surface, we confirmed a potential asymmetry in saccade-related burst strength between upper and lower visual field saccades[23].

A potential concern related to the above interpretation might be the curvature associated with the SC's three-dimensional shape. Since all electrode paths were constant and defined by the recording chamber's orientation ("Methods"), it could still be possible that lateral recording sites (representing the lower visual field) could have had systematically different depths from the SC surface than medial recording sites (representing the upper visual field), by virtue of the different SC surface curvature at the two groups of sites. We, therefore, decided to analyze a stricter grouping of SC neurons. We picked a smaller range of eccentricities (5–15 deg), directions from the horizontal meridian (<30 deg), and depths from the SC surface (1100–1900 μm) for comparing upper and lower visual field neurons' motor bursts. Our prior work on SC surface topography and 3-dimensional anatomical shape[33] suggested that this range of selection should reduce potential systematic differences in estimates of depths from the SC surface between the upper and lower visual field groups of neurons. We found 31 neurons (20 upper visual field and 11 lower visual field) satisfying the above strict criteria. When we analyzed their saccade-related firing rates, we still found a similar asymmetry between upper and lower visual field locations (Fig. 1c). The average firing rate (in our same measurement

interval) for the upper visual field neurons was 92 spikes/s, and it was 147 spikes/s for the lower visual field neurons; this difference was, again, statistically significant ($p = 0.0162$, t-test, t-statistic: −2.5524, df: 29). Therefore, it is likely, given both analyses in Fig. 1, that there is indeed a systematic asymmetry in SC motor burst strength between saccades towards the upper and lower visual fields. We were now in a position to ask whether such an asymmetry was reflected in saccade kinematics, as might be predicted from some recent as well as classic models of the SC rate code[13,14,16].

**Similarity of movement kinematics for saccades towards the upper and lower visual fields**. According to existing models, which assume directional symmetry in the SC movement commands, amplitude-matched saccades to the upper and lower visual fields should have different movement kinematics given the different SC motor burst strengths that we saw in Fig. 1. To test this, we analyzed saccades from both monkeys (P and N) from the same delayed, visually-guided saccade task that was used to analyze the peri-saccadic SC firing rates above ("Methods"). To compare size- and direction-matched movements, we picked, in each monkey, 5 saccade sizes (3, 5, 7, 10, and 13 deg radial amplitude), and two example directions from the horizontal meridian (+45 and −45 deg; i.e., oblique saccades; note that we also made similar observations for example directions that were nearer to or farther away from the horizontal meridian than +/− 45 deg). For each of the saccade sizes, we picked movements landing within a radius of 0.5, 0.8, 1, 2, and 3 deg, respectively, for the increasing saccade amplitude categories listed above. Therefore, we ensured that the movement endpoints were matched for landing accuracy. Example such movements are shown in Fig. 2a. In this figure, we only plotted rightward movements in monkey N and leftward movements in monkey P, for simplicity, but Fig. 2b shows both rightward and leftward saccades in each of the two monkeys. As can be seen from Fig. 2a, there was no clear difference in the trajectories of upward (light blue) versus downward (light red) oblique saccades, despite the significant SC neural asymmetry in Fig. 1. In fact, the pink upward traces in Fig. 2a are the same as the light red downward traces in the figure, but now reflected across the horizontal meridian for easier comparison to the upward saccades shown in light blue. These pink traces clearly overlapped strongly with the upward saccades.

Across the population of measurements from the above saccades, we plotted radial eye speed as a function of saccade amplitude and direction (Fig. 2b). This kind of plot summarizes the kinematics of the eye movements[34,35]. For each saccade size and right/left direction in each monkey, we plotted the radial eye speed for either upward (light blue) or downward (light red) oblique saccades (error bars denote 95% confidence intervals). There were no systematic differences in the saccadic profiles of the two groups of movements (across all sizes tested), despite the systematically stronger SC motor bursts for downward saccades seen in Fig. 1 (compare light blue and light red profiles for each saccade size). For example, stronger motor bursts in Fig. 1 could have predicted systematically higher peak speeds for the saccades directed towards the lower visual field[16]. This was clearly not the case (Fig. 2b). In fact, lower visual field neurons possess larger movement fields than upper visual field neurons[23], which should further increase the number of active spikes during saccade-related bursting for saccades towards the lower visual field; nonetheless, the kinematics of the movements were largely the same as those of upper visual field saccades (Fig. 2). Therefore, the results so far are consistent with a dissociation between SC saccade-related motor burst strength (Fig. 1) and saccade kinematics (Fig. 2).

**Similarity of upper and lower visual field saccade kinematics for a variety of behavioral contexts**. To further assess the dissociation between upper/lower visual field SC motor burst asymmetries and upward/downward saccade kinematics, we next turned to another, larger database of saccades for analyzing kinematics in more detail (database 2; "Methods"). In this case, we used: (1) immediate, visually-guided, (2) delayed, visually-guided, and (3) memory-guided saccades of different sizes and directions, with the sizes ranging from those associated with fixational microsaccades (approximately 0.1–0.2 deg) to approximately 15–20 deg. Aspects of these movements were analyzed previously for other purposes than movement kinematics[36,37]. Here, we wanted to confirm that the results of Fig. 2 still held for a larger range of movement amplitudes and directions, and also under different behavioral contexts. In other words, we analyzed the movements' kinematic properties in database 2, properties which were not analyzed in the prior publications. Moreover, database 2 allowed us to include data from a third monkey, M, when assessing potential differences (or lack thereof) in saccade kinematics between the upper and lower visual fields.

In our recent work with this database[36], we reported that saccadic reaction times were systematically shorter for upper visual field target locations when compared to lower visual field target locations, consistent with the asymmetry of SC visual neural sensitivity[23], and also consistent with other behavioral evidence[38–41]. For example, in Fig. 3a, we plotted example oblique saccades from monkey N from this database (but in a format similar to that used in plotting the data of Fig. 2a). In Fig. 3b (left), we plotted the absolute value of vertical eye position from the saccades shown in Fig. 3a (to facilitate comparing the upward and downward movements). Here, we temporally aligned the movements to the time of the go signal for triggering the saccades (the peripheral targets were continuously visible). Even though the saccade trajectories looked similar in Fig. 3a (save for the upward and downward distinction), the reaction times of the movements were markedly different (Fig. 3b, left). Saccades towards the upper visual field were triggered significantly earlier than saccades towards the lower visual field[36]. We then replotted the same saccades, but this time by aligning them to the time of peak radial eye speed during the movements (Fig. 3b, right). The movement kinematics were largely overlapping, with similar acceleration and deceleration profiles.

To summarize this kinematic similarity across all saccades in this database, we generated plots of peak eye speed as a function of saccade amplitude for each monkey[34,35]. In the first row of Fig. 4, these plots were made for the immediate, visually-guided saccade task, in which the fixation spot was extinguished at the same time as the appearance of the eccentric stimulus ("Methods"; this task was not used during SC recordings because the visual and motor bursts would occur too close to each other for proper neural analysis). For both monkeys N and M, there was very minimal difference in the main sequence relationship between saccades towards the upper (light blue) or lower (light red) visual fields, and any difference was certainly much smaller than the neural effects in Fig. 1. In fact, the insets in the first row of Fig. 4 show the reaction time results for the very same saccades, which are replicated from our recent work[36] for clarity. Despite a large effect of the visual field location on the movements' reaction times (also seen in Fig. 3b, left), there was minimal difference in saccade kinematics. This is again supportive of a dissociation between saccade-related SC motor burst strengths (Fig. 1) and movement kinematics (Figs. 2, 3); also see Figs. 6–8 below.

With an even larger database of visually-guided movements, now from the delayed, visually-guided saccade task, the same conclusion could be reached: the middle row of Fig. 4 shows

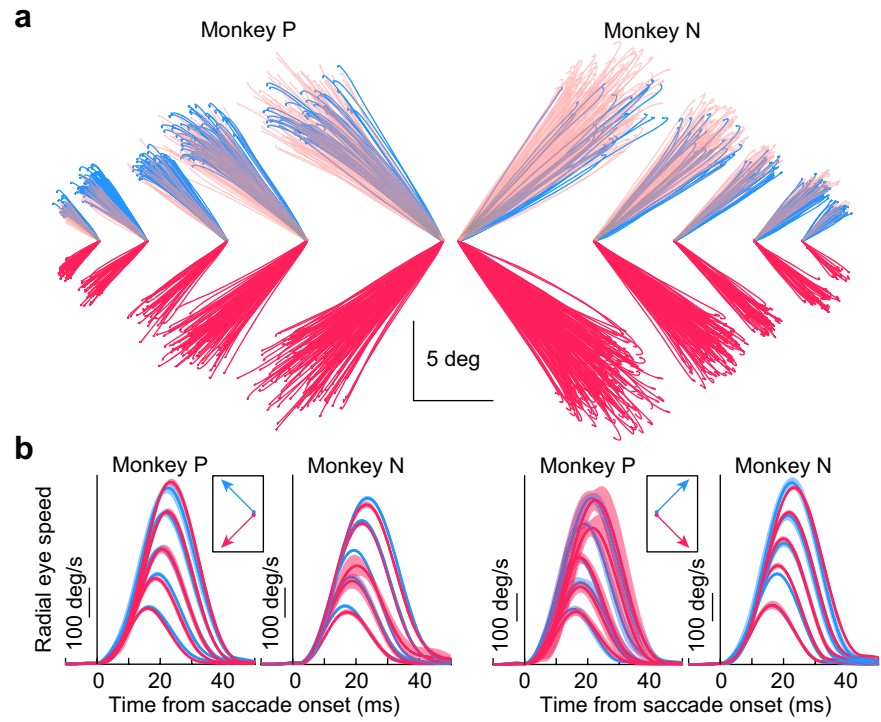

**Fig. 2 Upward and downward oblique saccades from the same sessions as the neural recordings exhibited similar kinematic properties, despite the neural asymmetry. a** Oblique saccades of different directions and amplitudes in both monkeys (rightward for N and leftward for P). Each line plots the horizontal and vertical displacement of eye position for a given saccade having a +45 deg (light blue) or −45 deg (light red) direction from the horizontal meridian. Saccades of similar sizes are grouped together (to start from the same origin) and displaced horizontally in the figure from saccades of different size ranges (the scale bars apply to all sizes). The pink traces overlaying the light blue traces are the same as the light red traces of the lower visual field saccades, but now reflected along the vertical dimension for better comparison to the upper visual field saccades. The trajectories of the saccades were largely similar regardless of direction from the horizontal meridian, despite the asymmetry in motor bursts in Fig. 1. **b** For each monkey, we plotted radial eye speed as a function of time from saccade onset for leftward (left pair of plots) or rightward (right pair of plots) saccades. In each case, we separated upward and downward movements by color (as in **a**). The different saccade sizes in **a** are reflected in the different peak speeds[34, 35]. The kinematic time courses of saccade acceleration, peak speed, and deceleration were largely similar for upward and downward saccades, despite the asymmetry of SC motor bursts in Fig. 1. Error bars denote 95% confidence intervals. The numbers of trials per condition in monkey N ranged from 7 to 362 (mean 121.2), and the numbers of trials per condition in monkey P ranged from 5 to 252 (mean 72.25).

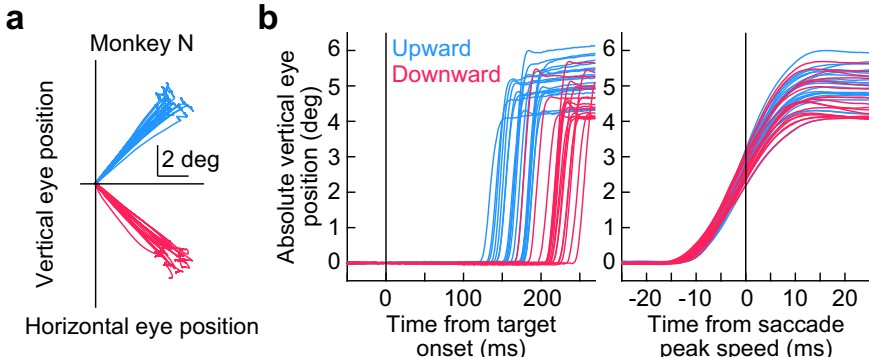

**Fig. 3 Upward and downward saccades exhibit strong differences in reaction times, but not kinematics. a** From the second database, we plotted, by way of example, similarly-sized oblique saccades from one monkey, and we separated them as being either upward or downward (as in Fig. 2a). **b** Relative to stimulus onset (left panel), the saccades were very different from each other in terms of their reaction times, as we characterized in detail earlier[36]; saccades towards the upper visual field (light blue) had significantly shorter reaction times than saccades towards the lower visual field (light red). However, kinematically, the saccades were very similar when aligned to the time of peak intra-saccadic eye speed (right panel). In both panels, we plotted the absolute value of vertical eye position displacement for each saccade (for better comparison of the upward and downward movements). Figures 4, 5 summarize the saccade kinematic results that we obtained for a much larger number of movements, and for two monkeys. $N = 21$ upward saccades and $N = 18$ downward saccades in this figure.

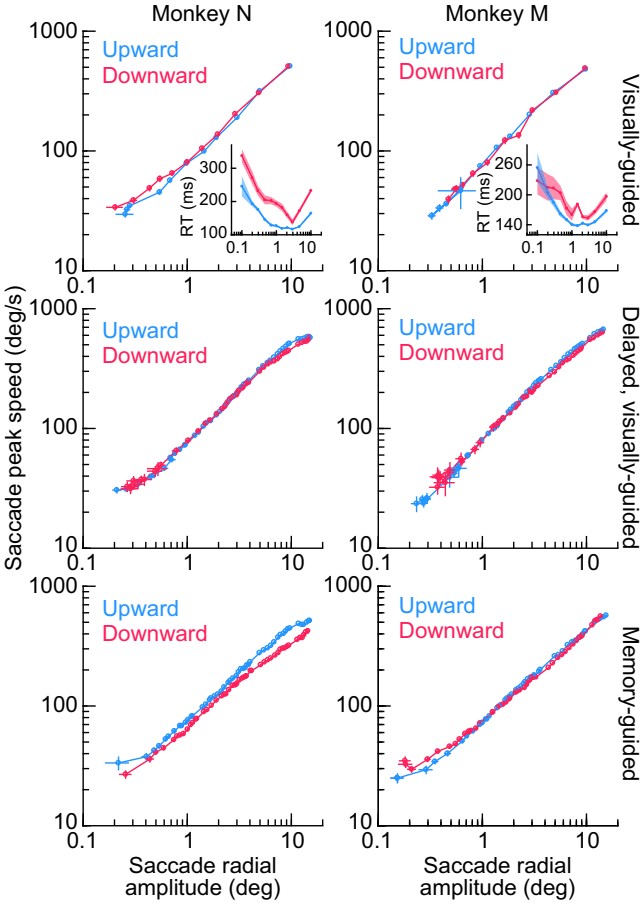

**Fig. 4 The main sequence relationship between peak eye speed and saccade amplitude does not depend on whether saccades are upward or downward for a variety of behavioral task contexts.** In each monkey (left and right columns), we plotted the main sequence from database 2 but after first separating saccades as being directed towards either the upper visual field (light blue) or lower visual field (light red). The first two rows show visually-guided saccades (immediate for the first row, and delayed based on a task instruction in the second row; "Methods"). The third row shows memory-guided saccades towards a blank region of the display. The insets in the first row show saccadic reaction time (RT) data (as in Fig. 3b, left) for the same saccades as in the main sequence plots, to highlight the strong presence of a visual field effect on reaction times and a concomitant absence of a visual field effect on saccade kinematics. In all saccade contexts (across rows), the visual field location of the saccade endpoint had minimal effect on saccade kinematics (despite a large effect on saccadic reaction times and despite an asymmetry in SC motor bursts; Fig. 1). The insets in the first row directly replicate the plots in Fig. 4a, c of ref. [36] for easier comparison of effect sizes for reaction times and kinematics. Error bars denote s.e.m. Note that monkey N showed a small reduction of peak eye speed for downward saccades when compared to upward saccades only in the memory-guided condition (bottom row), but this effect is opposite from what would be expected if SC motor bursts (Fig. 1) dictated kinematics. The insets were replotted with permission from ref. [36].

virtually no difference in the saccade kinematics between upward and downward visually-guided saccades, despite a clear effect size for SC motor-related neural responses in Fig. 1 and ref. [23]. The peak speeds in this row were also consistent with the peak speeds in the first row of Fig. 4 obtained with the immediate, visually-guided saccade task, as might be expected given the presence of a visual target for the saccades in both tasks.

We also tested memory-guided saccades. Even though such saccades were generally slower than visually-guided saccades (compare the bottom row of Fig. 4 to the two rows above it; also see Figs. 6–8 below)[24–30], the above-mentioned kinematic similarity between movements towards the upper and lower visual fields still persisted in the memory-guided saccade task. Only in monkey N (left column of the bottom row of Fig. 4) was there a reduction in downward saccade peak eye speed when compared to upward saccade peak eye speed. However, even in this case, such a reduction was inconsistent with the stronger saccade-related motor bursts for lower visual field saccades seen in the SC neural analyses (Fig. 1). If anything, stronger lower visual field SC motor bursts (along with larger response fields[23]) might predict higher, rather than lower, peak speeds for downward saccades.

Therefore, across a large range of movement sizes and directions, we found minimal kinematic differences between amplitude-matched upper and lower visual field saccades, even though other aspects of saccade generation (such as reaction times; insets in Fig. 4) were strongly different, and even though SC saccade-related motor bursts were also different (Fig. 1).

Finally, we also checked saccade durations as a function of saccade amplitudes (Fig. 5), and we reached similar conclusions. Saccade duration versus amplitude curves strongly overlapped for saccades towards the upper (light blue) and lower (light red) visual fields (Fig. 5), and this was true across task contexts. Note how monkey N compensated for the slightly slower downward memory-guided saccade peak speeds (when compared to upward memory-guided saccade peak speeds) with mildly longer durations for these movements (left column of the bottom row of Fig. 5). This might suggest that there was lower drive for generating this monkey's downward memory-guided saccades in general, which was then compensated for by increased movement durations. Nonetheless, as stated above, this is an opposite effect from what might be expected from the neural burst strengths in the SC (Fig. 1).

**Dissociation between SC motor burst strengths and movement kinematics also for vector-matched saccades.** In the above experiments, and as stated above, we were motivated by an assumption of directional isotropy in models of saccade control by the SC rate code[13,16,18]. In such models, saccades are implemented (in terms of efferent connection strengths towards the brainstem) according to their amplitude not vector; as a result, analyses of experimental data often collapse measurements across different directions. We reasoned that if this was indeed the case, then different SC burst strengths for upward and downward saccades (Fig. 1) should lead to different saccade kinematics, which we did not observe (Figs. 2–5). Having said that, it may be argued that our observations so far merely suggest a different efferent mapping to the downstream oculomotor control circuitry from the upper and lower visual field SC representations, rather than a dissociation between SC motor burst strengths and movement kinematics. While such a different efferent mapping between the upper and lower visual fields would indeed be interesting, we elected to further test our original hypothesis using a complementary approach, this time by employing vector-matched saccades of different kinematics.

We exploited the fact that saccades to a blank (as in memory-guided saccades) can be slower than visually-guided saccades[24–30]. We thus instructed 3 monkeys (M, N, and A) to perform delayed, visually-guided saccades and memory-guided saccades towards the response field hotspot locations of SC neurons. This meant that we now had even more SC recording data from monkey N (beyond those shown in Fig. 1), as well as

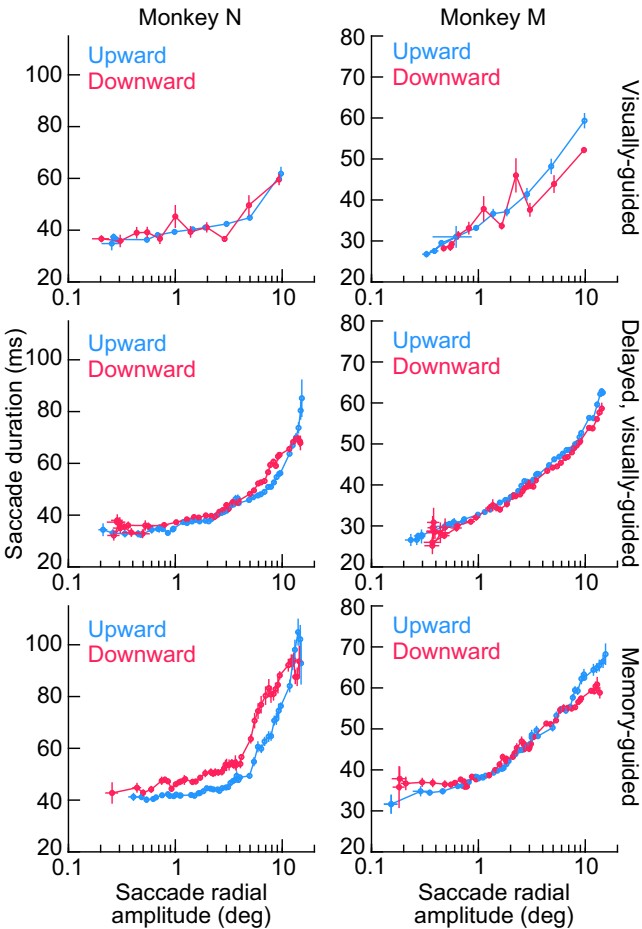

**Fig. 5 Saccade durations as a function of saccade amplitude are also largely insensitive to whether saccades are directed towards the upper or lower visual field.** Same as Fig. 4 but now plotting saccade duration as a function of saccade amplitude. Similar conclusions were reached concerning the minimal influence of upper versus lower visual field saccade target locations on saccade kinematics, despite asymmetries in SC motor bursts (Fig. 1). For monkey N in the memory-guided condition (left panel of the bottom row), the slower lower visual field saccade peak speeds (Fig. 4) meant slightly longer saccade durations when compared to upper visual field saccades (a speed-duration tradeoff). Error bars denote s.e.m.

additional SC motor burst measurements from two more monkeys (M and A), always comparing vector-matched visually-guided and memory-guided saccades towards hotspot locations. Moreover, in all 3 monkeys in this additional database (referred to here as database 3), we also recorded neurons using linear electrode arrays (monkey M also contributed some single-electrode sessions as well; "Methods"). This meant that we sometimes had simultaneously recorded neurons for the same behavioral trials. For each isolated neuron, we first selected saccades matched by direction and amplitude across the two tasks (data filtering procedures, and minimum trial count requirements, are detailed in "Methods"). We then checked the movement kinematics across the two tasks, and we evaluated how SC motor bursts were potentially modified.

Figure 6a shows an example saccade vector from one of our sessions. In purple, we show the delayed, visually-guided saccades from the session, and in green, we show the memory-guided saccades. Despite being matched in direction and amplitude (as per our experimental design), almost all memory-guided saccades from this session were slower than all visually-guided saccades, as can be seen from Fig. 6b. Thus, we had vector-matched saccades

with clearly differing kinematics. We then checked how the SC motor bursts were altered. In Fig. 6c, we show the spike waveforms of 3 different SC neurons that we recorded simultaneously from the same session in the two tasks. Each sub-plot in Fig. 6c shows the mean and standard deviation of a random sampling of spike waveforms from a given isolated neuron in both tasks ("Methods"). As can be seen, the waveforms were almost completely overlapping for each neuron, suggesting that isolation quality was sufficiently stable for each of them as we sequentially ran the two behavioral tasks ("Methods"). Therefore, we were now in a position to compare the motor bursts of the neurons in the two behavioral contexts.

Surprisingly, there was a large diversity of motor burst modulations between the visually-guided and memory-guided saccades, despite the highly consistent kinematic effects seen in Fig. 6b. For example, Neuron 1 in Fig. 6d had a weaker burst in the memory-guided saccade condition than in the visually-guided saccade condition, consistent with the kinematic effect across the two conditions. However, Neuron 2 was much less affected by the behavioral manipulation, and, most surprisingly, Neuron 3 had a much stronger motor burst in the memory-guided condition instead of the visually-guided condition (Fig. 6d). Thus, there was no systematic reduction in SC motor burst strengths (Fig. 6d) for the systematically slower (but vector-matched) memory-guided saccades (Fig. 6b), as would be predicted by current models of kinematic control by SC motor bursts.

The above observations were repeatedly seen across our experiments. For example, in Fig. 7, we plotted the results from another example SC site, this time in the SC's lower visual field representation. There was still a diversity of SC motor burst strength modulations as we went from the visually-guided to the memory-guided saccade paradigms (Fig. 7d), despite the matched saccade vectors (Fig. 7a), and also despite the clear kinematic differences between the two conditions (Fig. 7b). There was also, again, a neuron in this session (Neuron 6) that violated the clearly slower saccades observed in Fig. 7b for the memory-guided saccade condition. Thus, this additional example site revealed the very same patterns as those shown in Fig. 6, and it suggests a dissociation between SC motor bursts and saccade kinematics. A similar conclusion was also reached in earlier comparisons of the two tasks[24].

In total, we analyzed 114 SC neurons from 71 sites in these vector-matched experiments (from monkeys M, N, and A; "Methods"). To summarize their results, we first confirmed that all saccades were vector-matched across the visually-guided and memory-guided saccade conditions, as per our experimental design. For each of the 71 sites, each having a unique saccade vector, we collected the average saccade vector from each of the two conditions. We then plotted the amplitude (Fig. 8a) and direction (Fig. 8b) of the memory-guided saccade vector against the amplitude and direction of the visually-guided saccade vector. There was no difference between the two conditions in either amplitude (Fig. 8a) or direction (Fig. 8b) (amplitude comparison: $p = 0.5313$, paired t-test, t-statistic: $-0.6292$, df: 70; direction comparison: $p = 0.9735$, paired t-test, t-statistic: $-0.0334$, df: 70), as expected (we explicitly matched the vectors of the saccades in these experiments; "Methods"). We then confirmed that the saccades were significantly slower in the memory-guided condition than in the visually-guided condition[24-30], and we did so by plotting in Fig. 8c the peak speeds from all experiments against each other, in a fashion similar to the amplitude and direction plots of Fig. 8a, b. Memory-guided saccades were significantly slower than vector-matched visually-guided saccades ($p = 6.21 \times 10^{-13}$, paired t-test, t-statistic: $-8.7979$, df: 70). Most importantly, we then related the peak speed effect (for the vector-matched movements) to the neural motor burst effect. To do so,

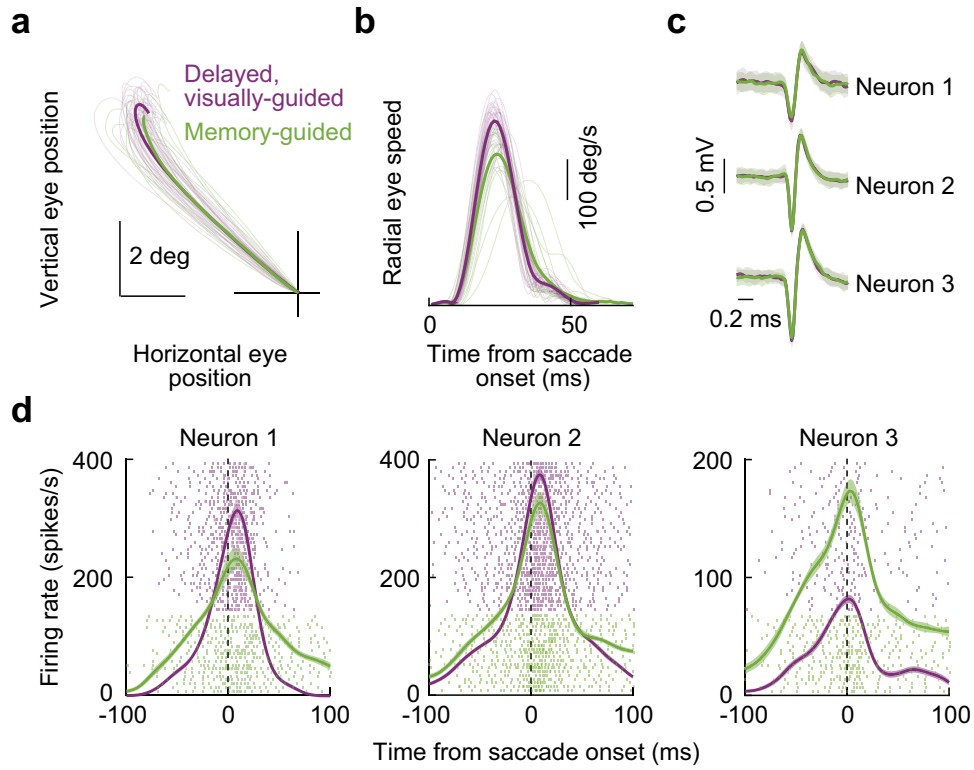

**Fig. 6 Superior colliculus motor burst strength is dissociated from saccade kinematics even for vector-matched movements. a** Example saccade trajectories from one session of the vector-matched experiments. Thin lines show individual movements, and thick lines show averages across trials. Purple denotes delayed, visually-guided saccades, and green denotes memory-guided saccades. The vectors of the two types of saccades were matched ("Methods"). **b** Memory-guided saccades were systematically slower than their visually-guided counterparts, as might be expected[24–30]. **c** Spike waveforms from 3 example neurons recorded simultaneously during the same session. Each thick line shows the average of a random sampling of waveforms from each of the two tasks. The error bars denote the s.d. across observations. In all cases, the waveforms were stable across the two task types, suggesting that each neuron was successfully recorded in the two tasks. The numbers of waveforms included in the averages for the visually-guided and memory-guided conditions, respectively, are: 70/97 (Neuron 1), 79/74 (Neuron 2), and 42/54 (Neuron 3). **d** The neurons of **c** had a large diversity of effects, in terms of peri-saccadic motor burst strengths, as a function of saccade type. Neuron 1 had a weaker motor burst strength for memory-guided saccades; Neuron 2 was less affected by the condition; and Neuron 3 had a surprisingly stronger motor burst in the memory-guided condition, despite the significantly slower saccades seen in **b**. Error bars denote s.e.m. and numbers of trials are evident from the shown individual-trial spike rasters.

we measured each neuron's peak firing rate in the interval −25 to +25 ms from saccade onset ("Methods"). We then created a neural modulation index as the burst strength in the memory-guided condition minus the burst strength in the visually-guided condition, divided by the sum of the two burst strengths ("Methods"). Values of the index >0 would indicate that motor bursts were actually stronger in the memory-guided saccade condition than in the visually-guided saccade condition. Similarly, we created a behavioral modulation index as the peak saccade speed in the memory-guided condition minus the peak saccade speed in the visually-guided condition, divided by the sum of the two peak speeds ("Methods"). Across all neurons, there was no correlation between the neural and behavioral modulation indices (Fig. 8d) (Pearson correlation coefficient: −0.1374, $p = 0.1449$). Rather, there was a more-or-less constant behavioral effect (slower saccades in the memory-guided condition) irrespective of SC neural modulation effect, as evidenced by the vertical scatter of points across all neurons in Fig. 8d.

Interestingly, there was a large dynamic range of neural modulation indices. Some neurons were almost completely suppressed in the memory-guided condition. These can qualify as visually-dependent saccade-related neurons[24,31,37]. Alternatively, 46.49% of the neurons (53/114) were above the diagonal line in Fig. 8d, thus violating the predictions of current models of kinematic control by SC motor bursts. Most intriguingly, almost

one quarter of the neurons (23.7%, 27/114) had a neural modulation index >0, suggesting that these neurons actually exhibited stronger motor bursts for memory-guided saccades than for vector-matched visually-guided saccades (Fig. 8d), despite the significantly slower speeds of the former (Fig. 8c). We observed such neurons, which tended to have weaker motor burst strengths in the visually-guided condition than the other neurons of our database, at all depths in our experiments. Thus, the motor bursts were independent of the actual triggered saccades.

These results, combined with those of Figs. 1–5 above, suggest that there is indeed a dissociation between SC motor burst strengths and saccade kinematics.

## Discussion
We described a dissociation between SC saccade-related motor burst strengths and movement kinematics. In particular, we confirmed an asymmetry in motor burst strengths between upper and lower visual field saccade target locations (Fig. 1). We then found that the kinematics of amplitude-matched saccades towards upper and lower visual field locations were not different from each other across a range of movement sizes, directions, and behavioral contexts (Figs. 2–5). Finally, we demonstrated how there was no correlation between SC motor burst effects and

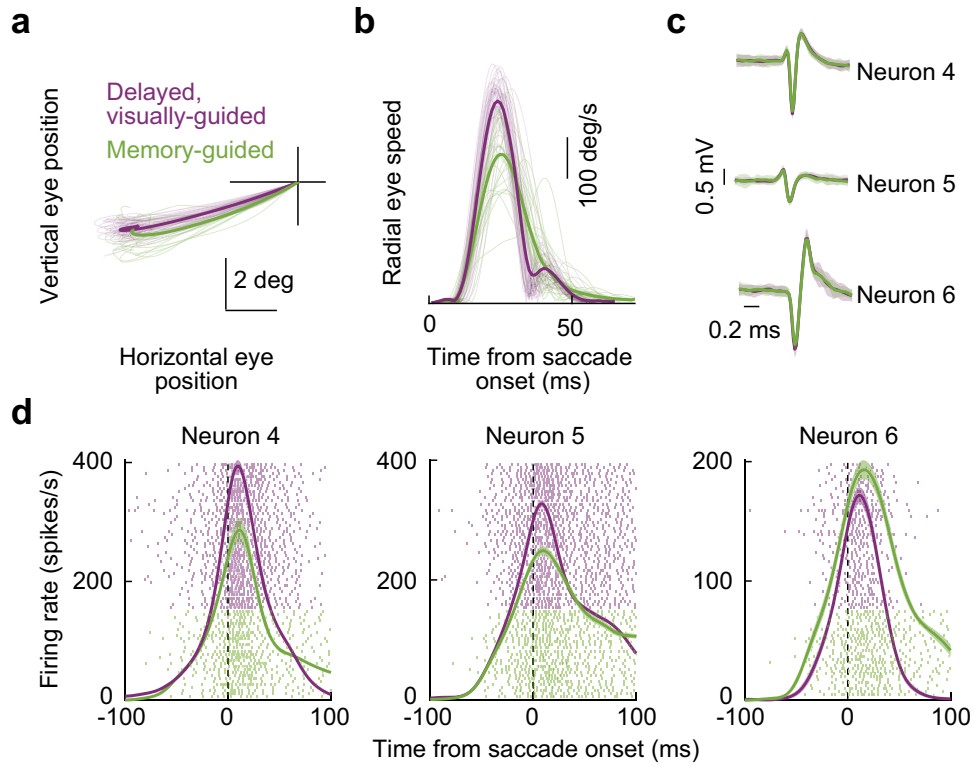

**Fig. 7 Similar observations from another example SC site.** The figure is formatted identically to Fig. 6. The site of this example session was now in the SC's lower visual field representation, as evidenced by the downward oblique saccades in (**a**). Note how the saccade speed was clearly different between visually-guided and memory-guided saccades (**b**), but the neurons still had a diversity of effects in terms of motor burst strengths (**d**). The numbers of spike waveforms included in the averages of **c** for the visually-guided and memory-guided conditions, respectively, are: 117/82 (Neuron 4), 76/65 (Neuron 5), and 85/81 (Neuron 6). Error bars in **c** denote s.d., and error bars in **d** denote s.e.m.

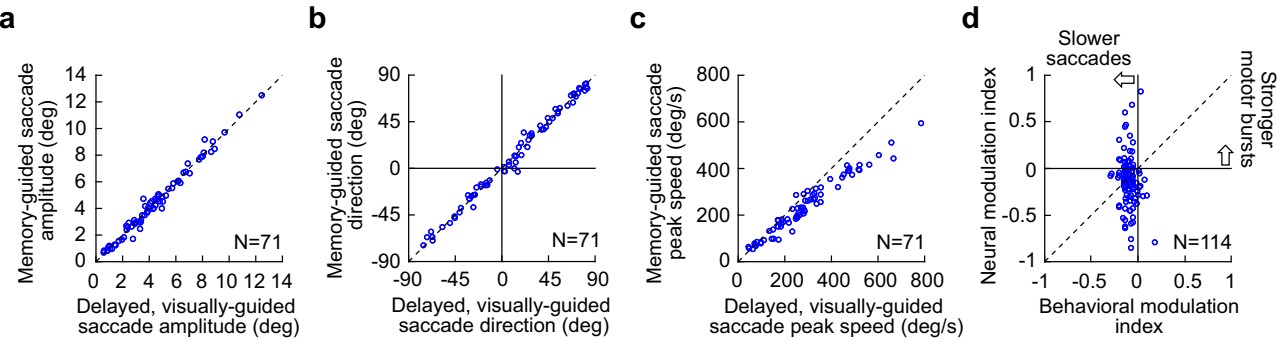

**Fig. 8 Population summary demonstrating how SC motor burst strength is dissociated from saccade kinematics even for vector-matched movements.**
**a** Average saccade amplitude in the memory-guided saccade task versus the visually-guided saccade task across all unique sessions in this experiment. Each symbol indicates a single session. As per the experimental design, the saccade amplitudes were matched across the two conditions. **b** Same as **a** but for the directions of the saccades. Negative means downward saccades, and positive means upward saccades. Again, there was no difference in saccade angles across the visually-guided and memory-guided conditions. Thus, **a** and **b** indicate that the saccades in the two conditions were vector-matched. **c** Despite the vector matching, peak speeds were consistently lower for memory-guided saccades. **d** For all recorded neurons from the same sessions, we calculated a behavioral modulation index, in which negative values indicated slower saccades in the memory-guided condition. We also created a neural modulation index, in which negative values indicated weaker motor bursts in the memory-guided condition. Note that some neurons were recorded simultaneously during the same trials, as in Figs. 6, 7 ("Methods"). Thus, there were more neurons than sessions, explaining the different numbers of symbols in this panel relative to **a**–**c**; this also meant that there could be multiple symbols with different y-axis values but having the same x-axis value (because multiple neurons were collected for the same behavioral trials). There was no correlation between neural and behavioral modulation indices.

kinematic alterations even in vector-matched saccades towards response field hotspot locations (Figs. 6–8).

Other examples of dissociations between SC motor burst strengths and movement properties are consistent with our interpretation that the SC rate code does not necessarily dictate movement kinematics, as might be suggested by some recent

models[13,16]. For example, and as we have confirmed (Fig. 8d), for memory-guided saccades towards response field hotspot locations, a significant fraction of SC neurons becomes silent at the time of movement onset[24,31,37]. This, in addition to neurons that exhibit potentially altered response field profiles when making saccades towards a blank[29], suggests a significantly modified SC

representation during these movements. Indeed, we even found neurons that exhibited stronger, rather than weaker, motor bursts for slower memory-guided saccades (Fig. 8). Therefore, the relationship between movement kinematics and SC motor bursts is relatively loose when making saccades towards a blank, and it was also relatively loose in our analyses of upper and lower visual field target locations.

Another example of a dissociation between saccade motor burst strength and movement kinematics was observed when saccades were driven by combinations of visual and auditory sensory signals, as opposed to only visual signals[17]. Interestingly, it was again the case in this example that a sensory scenario was relevant and critical for revealing a potential separation between the SC rate code and movement kinematics. That is, in both the example above of saccades towards a blank as well as the current example of multi-sensory target specification, it was a modification of a sensory property of saccade targets that has allowed observing a dissociation between motor burst strengths and eye movement properties. This clear context-dependence of the bursts indicates that SC motor bursts are likely not pure motor controllers, in the strictest sense of the word. In the current manuscript's context as well, we were originally motivated by the fact that it was visual sensitivity that was strongly variable between upper and lower visual field locations[23]. Indeed, given that stronger visual responses occur in the upper visual field whereas stronger motor responses occur in the lower visual field, it is intriguing to consider the possibility that there might be a general anti-correlation property between visual sensitivity and saccade-related motor burst strength in the SC, for example, in the ubiquitous visual-motor neurons of this structure.

More recently, Peel and colleagues also identified a dissociation between SC motor burst strength and executed saccade properties[22]. Specifically, these authors causally perturbed top-down inputs towards the SC through reversible cooling of the cortex, and they found reduced SC burst strengths for metrically similar executed saccades. This study, along with refs. [17,24,31,37], all showed that for the very same saccade vector (i.e., only within the upper visual field SC representation or only within the lower visual field SC representation), sensory[17,24,31,37] or physiological[22] manipulations can indeed significantly alter SC burst strengths without fully accounting for the altered saccade properties. Therefore, whether one considers a single saccade vector like in these studies or a comparison of upper versus lower visual field saccades like in the first half of our study, a dissociation between SC saccade-related motor bursts and saccade execution still exists.

Naturally, our comparison of upper and lower visual field saccades is somewhat different from the approach used to test existing models of the SC rate code[13,16]: these models have primarily focused on a single saccade vector, whereas we compared amplitude-matched saccades of different directions. We believe that such comparison has allowed us to learn something potentially very interesting about the transformation of desired saccade vector information into eye muscle innervations that has to take place in the brainstem downstream from the SC. Specifically, our results suggest that structures downstream from the SC might compensate for the SC asymmetry shown in Fig. 1 and in ref. [23], to result in similar eye movement kinematics for upward and downward eye movements. Even though we also explicitly performed single vector experiments (to response field hotspot locations; Figs. 6–8) and found results consistent with our original hypothesis, this approach of comparing saccades of different directions can be fruitful, in general, in research on the oculomotor system. Indeed, the large SC asymmetry in both saccade-related movement burst strengths (Fig. 1) as well as other visual and motor discharge properties[23] has real consequences for

saccade latencies, saccade accuracy, and the likelihood of express saccades[23]. Thus, the SC asymmetry motivates investigating what the functional role of SC motor bursts is, and in a more general framework than just one of controlling saccade kinematics.

Our results also provide complementary evidence to a phenomenon that we recently studied[21]. In that recent study, we altered saccade metrics and kinematics by strategically injecting visual bursts into the SC at spatial sites beyond the vector endpoints of the currently executed movements[21]. We found alterations in movement metrics and kinematics[20], which were lawfully related to the amounts of visual spikes that we injected onto the SC map around the time of movement triggering[21]. These results were consistent with the spatial code of the SC because the movement metric changes reflected the topographic locations of the injected spikes. However, critically, the movements' kinematic alterations occurred in the absence of strong alterations in the motor bursts themselves (for the neurons generating the originally planned saccades). This was surprising for a variety of reasons, including ideas related to lateral connectivity patterns in the SC, as we recently discussed[21]. However, it also represented an opportunity for us to explicitly ask, in the current study, whether or not the rate code was indeed as tightly linked to individual movement kinematics. Therefore, here, in the first half of the study (Figs. 1–5), we took the opposite approach from our recent study: we identified a situation in which the motor bursts were different from each other for two different sets of amplitude-matched saccades, and we showed that the saccade kinematics in the two groups of movements were the same (Figs. 1–5). Of course, we also considered the case in which the same saccade vector was made towards the response field hotspot location but with different kinematics (Figs. 6–8). The net result is that either with altered movements and minimally-altered movement commands[21], or with minimally-altered movements and significantly altered movement commands (Figs. 1–5), or with vector-matched movements of different kinematics and a diversity of SC motor burst effects (Figs. 6–8), there does indeed seem to be a clear dissociation between saccade kinematics and SC motor burst strengths.

Recent work has suggested that the SC can support high level perceptual and cognitive phenomena[42,43]. For example, the SC causally influences selective behaviors[44,45], and it even shapes object-related visual representations in the ventral visual processing stream[46]. This is in addition to established roles for the SC in target selection[47–49]. All of this evidence suggests the SC might occupy a functional level that is slightly more abstract than that of specifying individual movement kinematics, consistent with our results. Thus, it might suffice for the SC to specify desired movement metrics, via the spatial code, and also potentially contribute to the decision of when to trigger an eye movement, as recently suggested[11]. The rest can be handled by downstream oculomotor control structures. If this is indeed the case, then a critical and urgent question for research in the immediate future is: what is, ultimately, the functional role of the SC rate code in visual-motor behavior and perception? One possibility could be that it allows providing a differential gain signal for cortical visual processing. For example, it is known that visual perception[50–53] and attention[54,55] are better in the lower visual field under conditions of gaze fixation. However, peri-saccadic perceptual mislocalization performance is different for upward saccades[56]. Moreover, when we recently measured perceptual sensitivity in peri-saccadic intervals, at the time of saccadic suppression[57], we found such sensitivity to be better in the upper visual field instead —consistent with a stronger peri-saccadic suppression of visual sensitivity in the lower visual field[58]. If SC motor bursts contribute to saccadic suppression, perhaps via inhibitory projections to the frontal cortex[32,59,60], then a possible functional role for

stronger motor bursts in the SC's lower visual field representation could be to differentially modulate cortical visual processing at the time of saccades. It would be interesting to investigate this hypothesis in future studies.

## Methods

**Study design**. In this study, we described results from three different sets of experiments, referred to here as database 1, database 2, and database 3, respectively.

In database 1, we analyzed data from our previously published study[23]. Specifically, neural activity from the SC and saccadic behavior were recorded from two adult, male rhesus macaque monkeys (P and N)[23]. We analyzed both neural activity and behavior from that study, using a delayed, visually-guided saccade task.

In database 2, we analyzed saccadic behavior that was recorded from monkey N and a third adult, male monkey (M), again from a previously published experiment[36]; here, we analyzed additional behavioral parameters from that study that were not previously described. We also analyzed multiple behavioral tasks.

In database 3, we analyzed both saccadic behavior and SC neural activity from adult, male monkeys N, M, and A. The experiments consisted of either single-electrode recordings in monkey M or linear electrode array recordings in all 3 monkeys. The linear electrode array recordings in monkeys N and M were re-analyzed from a previous study[37], whereas the linear electrode array recordings from monkey A (aged 10 years and weighing 10 kg), as well as the single-electrode recordings in monkey M, were all newly-performed experiments not previously described in any other publication.

Thus, we had SC neural recordings from a total of 4 monkeys (M, N, A, and P) and behavior from a total of 3 monkeys (M, N, and P) in this study.

All experiments were approved by the Regierungspräsidium Tübingen, under licenses CIN3/13 and CIN4/19G, and they were in accordance with the German and European directives on the use of animals in research.

In what follows, we describe detailed methods relevant for the current work.

**Animal preparation**. For SC recording, a recording chamber was implanted centered on the midline in all 4 monkeys. The midline positioning of the chamber allowed recording from both the right and left SC in each animal. Magnetic resonance images (MRI's) obtained prior to the experiments aided in chamber implant alignment. We aimed for quasi-orthogonal electrode penetrations (relative to the SC curvature) at eccentricities we typically use in experiments (e.g., 5–15 deg).

Before receiving the chamber implants, the animals were also implanted with head-holding apparatuses and scleral search coils for eye tracking, as described earlier[61–63]. The scleral search coils allowed using the magnetic induction technique for measuring eye positions[64,65]. Specifically, a coil of wire was implanted around the sclera of the eye and below the conjunctiva. The animals were then seated near the middle of a cube in which alternating magnetic fields induced electrical current (which depended on ocular position) in the implanted scleral coil; we measured and calibrated this electrical current.

**Behavioral tasks**. For both neural and behavioral analyses, the monkeys performed classic saccade generation tasks.

In the immediate, visually-guided saccade task of database 2 (which was only used in behavioral experiments and not neurophysiological experiments), the monkey first fixated a central spot. After a variable delay, the spot was jumped to another location, and a saccade to follow the spot was triggered.

In the delayed version of the same task, during initial fixation, the fixation spot remained visible while an eccentric spot was presented. The monkey was required to maintain gaze fixation and withhold any reflexive orienting towards the eccentric spot for as long as the central fixation spot was visible. After the fixation spot was removed, the monkey generated a saccade towards the (still visible) eccentric spot.

Finally, in the memory-guided saccade task, during initial fixation, the eccentric spot was only flashed briefly (for approximately 50 ms). A delay period then ensued in which only the fixation spot was visible, and the monkey was required to maintain gaze fixation on it. At the end of this so-called memory period, the fixation spot was extinguished, instructing the monkey to generate a saccade towards the remembered location of the previous flash (i.e., towards a blank location on the display).

The delayed, visually-guided saccade task was used for all neural analyses reported in this study (database 1 and database 3). This was important because this task allows dissociating visual burst intervals from the saccade-related motor burst intervals that we were interested in analyzing. The memory-guided saccade task was also used for neural analyses in database 3. For behavioral analyses, we used the delayed saccade task in database 1 (e.g., Fig. 2), as well as all 3 saccade tasks in database 2.

For mapping response fields, we generally employed the delayed, visually-guided, and memory-guided saccade tasks. However, in the newly-acquired portions of database 3 (monkey M single-electrode recordings and monkey A linear electrode array recordings), we first mapped response fields with a fixation variant of the delayed, visually-guided saccade task. That is, at the end of the trial, instead of fixation spot removal to release a visually-guided saccade towards the

eccentric target, the monkey was simply rewarded for fixating until trial end. This allowed us to obtain visual response fields, and we then later tested for saccade-related bursts using the delayed, visually-guided and memory-guided saccade tasks.

In all cases, stimuli were presented on cathode ray tube (CRT) displays, with stimulus luminances and dimensions having been described earlier[23,36,66]. The timing of trial events in the tasks was also described earlier. For the present study, the primary focus was on the individual saccade kinematics at the ends of all trials, irrespective of timing parameters, such as the length of the delay or memory period, and irrespective of the exact stimulus visual properties. The effects of these factors (such as trial timing or visual stimulus properties) were described earlier[23,36,67,68].

**Behavioral data analyses**. All saccades from databases 1 and 2 were detected for the previous two studies[23,36]. Here, we analyzed the kinematic properties of the movements. For database 3, the saccades from the electrode array recordings of monkeys M and N were also detected previously[37]. The saccades from the newly-acquired monkey M and monkey A recordings were detected using our standard approaches[61,69].

For behavioral analyses in database 1, we picked saccades having +45 or −45 deg direction from the horizontal meridian (i.e., oblique saccades). We then picked 5 radial amplitude categories to characterize 5 different ranges of saccade sizes (Fig. 2). The categories were: 3, 5, 7, 10, and 13 deg. For each of these categories, we picked all saccades landing within a radius of 0.5, 0.8, 1, 2, and 3 deg from the designated amplitude/direction category, respectively. For example, for saccades of 7 deg amplitude and +45 deg direction, we picked all saccades that were upward and oblique, and that were directed towards an eccentricity of 7 deg, and that landed within a radius of 1 deg from this eccentricity. Similarly, for 3 deg saccades of +45 deg direction, we picked all upward oblique movements towards an eccentricity of 3 deg and landing within a radius of 0.5 deg from it. This meant that we had amplitude- and direction-matched saccades for either the oblique upward or the oblique downward movements. We then plotted the trajectories (Fig. 2a) and radial speed profiles (Fig. 2b) of all of these saccades. Since the speeds of temporally-directed saccades could be different from the speeds of nasally-directed saccades for a given tracked eye, we analyzed rightward and leftward saccades separately in this database (Fig. 2b). However, in database 2, all saccade directions were combined, and with similar conclusions.

For database 2, we had a large range of saccade amplitudes and directions to analyze[36]. We plotted the main sequence relationship[34,35] for these saccades after separating them into two groups: saccades towards the upper visual field and saccades towards the lower visual field. We plotted both the main sequence relationship of peak speed versus movement amplitude (Fig. 4) and saccade duration versus movement amplitude (Fig. 5). For comparison, we included a plot of saccadic reaction times for the same saccades in Fig. 4. This was a replotting of the reaction time data already reported earlier[36], and we included it here for easier comparison of the difference in effects of visual field location on saccade kinematics and saccade reaction times. In total, we analyzed 1246, 928 visually-guided saccade trials, 6147, 5871 delayed, visually-guided saccade trials, and 6428, 9631 memory-guided saccade trials from monkeys N and M, respectively. The numbers of trials for the behavioral analyses from database 1 are reported in the figure legend of Fig. 2.

For database 3, our behavioral analyses consisted of first ensuring vector matching and then checking the movement kinematics to set the stage for neural data analyses. For the previously collected data[37] (monkey M and monkey N linear electrode array recordings), response fields were mapped with both the delayed, visually-guided and memory-guided saccade tasks. Therefore, in offline analyses, we obtained the average firing rate in the interval of −25 ms to +25 ms from saccade onset for each movement. We then plotted heat maps of firing rate as a function of saccade horizontal and vertical amplitudes to confirm the response fields. We identified the hotspot location of each neuron from the visually-guided saccade response field, and we then picked all saccades in both tasks landing within 2 deg, 1 deg, or 0.5 deg of this location depending on the neuron's preferred eccentricity (within 2 deg for neurons with preferred eccentricity > 3 deg, within 1 deg for neurons in the range of 2–3 deg preferred eccentricity, and within 0.5 deg for foveal neurons). We only included neurons for which we had at least 5 vector-matched saccades in each of the visually-guided and memory-guided saccade conditions.

For the newly-collected measurements of database 3 (monkey M single-electrode recordings and monkey A linear electrode array recordings), after mapping visual response fields with the fixation task (described above), we ran delayed, visually-guided saccades and memory-guided saccades towards the hotspot location (as assessed online during the experiment), collecting at least 20 trials per task. We then checked for endpoint matching. We found the median landing position from the delayed, visually-guided saccade task. Then, we only included saccades in both tasks that landed within 1 deg from this position. Once again, we only included neurons for which we had at least 5 vector-matched saccades in each of the visually-guided and memory-guided saccade conditions (typically much more).

After finding vector-matched saccades in database 3, we then proceeded to plot radial eye speed for the delayed, visually-guided, and memory-guided saccade tasks. We also collected measurements per session as follows: average saccade amplitude,

average saccade direction, and average saccade peak speed. This allowed us to plot these parameters across the two tasks (e.g., Fig. 8a–c), to confirm vector matching as well as to confirm different saccade kinematics across the two conditions.

To obtain a single behavioral modulation index across the two tasks in database 3, we measured, in each session, the average peak saccade speed in the memory-guided condition and the average peak saccade speed in the delayed, visually-guided condition. We subtracted the latter from the former, and then divided by the sum of the two. This gave us an index that ranged in values from −1 to +1, with indices >0 indicating that peak saccade speed was higher in the memory-guided condition and indices <0 indicating that peak saccade speed was higher in the delayed, visually-guided saccade condition. This gave us a single number that we could relate to a similar single number for SC motor burst strength modulation by the behavioral task (as described later below).

**Database 1 neural data analyses (Fig. 1)**. We analyzed peri-saccadic firing rates, as we did previously[23]. We obtained firing rates by convolving individual spike times with a gaussian kernel of 10 ms σ. For each neuron in the database of the previous study (containing > 400 neurons), we had identified (for saccade-related neurons) the saccades towards the neuron's preferred movement-related response field location (i.e., the locations for which the neuron's saccade-related bursts were the strongest). In the present study, we analyzed the firing rates for these preferred saccades. However, we constrained the choice of neurons according to the needs of the current study. Specifically, besides only considering extra-foveal neurons with saccade-related bursts, we matched neural depths between neurons from the upper and lower visual field representations of the SC (e.g., Fig. 1a). Specifically, since saccade-related motor bursts in the SC can vary in strength as a function of depth of the neurons from the SC surface[6], we only compared motor bursts after selecting neurons from the upper and lower visual field representations that had matched depths.

To do so, we first considered all neurons in the upper and lower visual field representations having a depth of 600–1850 μm from the SC surface. This range of depths is consistent with known depths of saccade-related activity in the SC[6]. Importantly, for the present purposes, this range of depths contained clear overlap between neurons in the upper and lower visual field SC representations (Fig. 1a). This allowed comparing the strengths of motor bursts between the selected depth-matched neurons. The resulting neural database had 136 neurons (Fig. 1).

To further confirm that there was no confound of neural depth from the SC surface in interpreting a visual field asymmetry in motor burst strength, we were concerned that the curvature of the SC surface could introduce systematic biases in depths of upper versus lower visual field neurons from the SC surface. For example, it could potentially be the case that the three-dimensional SC surface curvature combined with a constant electrode approach angle dictated by the recording chamber might systematically skew depth estimates: medial (upper visual field) electrode locations might potentially have depth estimates that could be systematically different from lateral (lower visual field) electrode locations in the chamber. This could simply be a function of whether or not a given electrode track was more or less perpendicular to the local SC surface topography at a given site. In our second analysis of neural activity, we, therefore, picked a range of electrode locations in which we expected minimal changes in SC curvature between upper and lower visual field representations. For example, mapping the SC surface topography on the anatomical SC[33] might suggest a similar relationship between electrode angle and SC surface for upper and lower visual field representations near the horizontal meridian and within a specific range of movement amplitudes. We therefore specifically picked neurons with movement-related response field hotspots near the horizontal meridian (within 30 deg direction in either the upper or lower visual fields) and with radial eccentricities of only 5–15 deg. We also picked a narrower depth of neurons for the comparison (1100–1900 μm from SC surface). With this stricter neural database (31 neurons), we again plotted peri-saccadic firing rates for neurons in the upper and lower visual field representations (Fig. 1c).

In all cases, a neuron was considered to be part of the upper or lower visual field representation if its preferred saccade (i.e., the movement-related response field hotspot location) was in the upper or lower visual field, respectively. This was also consistent with the known SC topographic representation[1,3,23,33], and it was already done in our previous study[23].

To statistically compare saccade-related activity strength between the upper and lower visual field representations in the SC, we measured the average firing rate in the final 50 ms before saccade onset for each neuron. We then statistically compared the firing rates of all neurons having movement-related response field hotspot locations in the upper visual field to the firing rates of all neurons having response field hotspot locations in the lower visual field (using t-tests). Note that measuring average firing rates is equivalent to counting spikes, which has been the standard method to analyze the rate code[13,18]. Also note that in our analyses of database 3 recordings (described below), we also picked a peri-movement burst measurement (that is, including epochs also after saccade onset) rather than only a pre-movement measurement as in database 1, with similar conclusions to this database's results.

**Database 3 neural data analyses (Figs. 6-8)**. We collected 25 sessions of linear electrode array recordings in monkey N, 16 sessions of linear electrode array recordings and 32 sessions of single-electrode recordings in monkey M, and 12 sessions of linear electrode array recordings in monkey A (all array recordings were performed with V-probes from Plexon, Inc.). The linear electrode array

recordings from monkeys N and M were a subset of those described for a previous study[37].

We used offline sorting to identify single neurons. For the single-electrode recordings, we used Plexon's Offline Sorter utility. The visually-guided and memory-guided saccade tasks were collected (in sequence) together in the same file, and we sorted both tasks together. For the linear electrode array recordings, we performed offline sorting using Kilosort[70], followed by manual curation using the phy software. For the data from ref. [37], we used the same sorting results that were obtained for the original study. Isolated neurons that exceeded an estimated false positive rate (ISI violation) of 10% or had an isolation distance below 30 were excluded from further analysis. We sorted linear electrode array recording data from an entire session simultaneously, thus tracking neurons across the different tasks that we ran (typically much more than the visually-guided and memory-guided saccade tasks). To check isolation stability (e.g., for Fig. 6c), we collected 2000 spike waveforms selected randomly from the same session. For our two tasks of interest, we took the waveforms from this sampling of waveforms that happened to come from either of the two tasks, and we plotted their distributions. Because the two tasks were typically run in succession, it was usually very likely that isolation was stable throughout both of them. This was also the case in the single-electrode recordings.

For each of the vector-matched saccades, we defined a motor interval as the 50 ms interval centered on saccade onset (that is, the interval spanning −25 to +25 ms from saccade onset). We also defined a baseline interval as the final 50 ms interval before stimulus onset (at trial beginning). We then statistically compared the baseline interval firing rate to the motor interval firing rate using either a t-test (for all the newly-collected data) or a ranksum test (for the data from ref. [37]). The choice of test was dictated by the fact that the old data had fewer numbers of trials because we selected saccades from response field mapping data, whereas in the newly-collected sessions, we explicitly collected repeated saccades from the same response field hotspot location. We only included a neuron if it had a significantly elevated average firing rate in the motor interval relative to the average firing rate in the baseline interval ($p < 0.05$) in either the delayed, visually-guided saccade task or the memory-guided saccade task or both.

To check for changes in SC motor bursts across visually-guided and memory-guided saccades, we calculated the peak firing rate in the motor interval (−25 to +25 ms from saccade onset) in each condition. We then calculated a neural modulation index similar to how we computed the behavioral modulation index above. That is, we subtracted peak firing rate in the visually-guided saccades from peak firing rate in the memory-guided saccades, and we divided by the sum of peak firing rates. Thus, a neural modulation index >0 indicated stronger SC motor bursts for memory-guided than visually-guided saccades.

**Statistics and reproducibility**. We analyzed SC recording data from 4 different monkeys, with consistent results. Similarly, we analyzed behavioral measurements from 3 monkeys. These numbers of animals increase confidence in the generalizability of the results, especially given how some observations were highly consistent with a large literature on SC saccade-related bursts (e.g., the depth profile shown in Fig. 1a).

For neural analyses, we statistically compared upper and lower visual field motor bursts on a neuron per neuron basis (Fig. 1). Similarly, in Figs. 6–8, we compared burst strengths on a neuron per neuron basis. In all cases, we collected a large enough sample of neurons to increase statistical confidence in the observations.

For eye movement analyses, we employed minimum count requirements (e.g., for the vector matching in the experiments of Figs. 6–8) to ensure enough replicates. Similarly, we used large numbers of replicates in the behavioral measurements of Figs. 2–5.

All relevant statistical tests are indicated in the figure legends or the associated Results text. Also, numbers of observations are indicated in the figures, in the figure legends, in "Results", or in the "Methods" text.

**Reporting summary**. Further information on research design is available in the Nature Portfolio Reporting Summary linked to this article.

**Data availability**
The databases generated and/or analyzed during the current study are available from the corresponding author on reasonable request. The source data for all plots are also shown in Supplementary Data 1 and 2.

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

## Acknowledgements

We were funded by the Deutsche Forschungsgemeinschaft (DFG) through Research Unit: FOR1847 (project: A6: HA6749/2-1). The following DFG-funded projects also contributed to the current study: HA6749/3-1, HA6749/4-1, and SFB1233, Robust Vision: Inference Principles and Neural Mechanisms (TP 11, project number: 276693517).

## Author contributions

Z.M.H. collected and analyzed databases 1 and 2. T.Z., T.M., M.P.B., and Z.M.H. collected and analyzed database 3. All authors wrote the manuscript.

## Funding

## Competing interests

The authors declare no competing interests.
