## [Peer Review File · Communications Biology]

Reviewers' comments:

Reviewer #1 (Remarks to the Author):

In this article by Ziad Hafed, the author systematically compared the saccade-related activity of the superior colliculus (SC) neurons, and kinematics of eye movements between the saccades directed to the upward and downward directions. It was found that the presaccadic bursts were stronger for downward saccades (interestingly, it was opposite to the visual responses in his previous study (Hafed and Chen 2016)), even though the saccade kinematics was very similar. Based on these findings, the author concluded that the SC motor bursts do not dictate the movement kinematics as indicated in the title of this paper.

Similar conclusion has been suggested in previous studies on metrically similar executed saccades. In this study, the authors made comparison between the two different saccade vectors and obtained the evidence which complements the proposals made by preceding studies. The analysis is based on a big database of the neuronal and behavioral data and the results are convincing.

Having said so, I have some problems in the conclusions of this article, one on conceptual aspect, and the other on the data presentation.

First, as I first read the title of this paper, I imagined that this study is on the comparison between saccade related activity and saccade kinematics for the same saccades as in preceding other studies. And as I read the content of this article, I felt somewhat awkward because the direct conclusion drawn from this article should be that the input-output gain of the upward and downward premotor saccade generator circuits downstream of the SC (presumably in the meso-diencephalic junction?) is different. Interestingly, this would raise another interesting question that the gain of horizontal saccade generator should be different depending on whether the saccade has the upward or downward component (this is just a comment). I think that the present paper supports the concept that the saccade kinematics are not solely determined by the SC, but primarily by the premotor circuits. Therefore, I think the present title is a bit too strong and general, for the conclusion directly drawn from this article by itself.

Second, the author described that the kinematics of delayed and memory-guided saccades are also the same between the upward and downward saccades. But if the author emphasizes that, he should show the neuronal firing data also for them. This is particularly important, because the Fig.1 is on the step visually guided saccades, where the visual and motor bursts are not completely dissociated. Therefore, to draw clear conclusion on the "motor bursts", the authors should show the data on the delayed or memory-guided saccades where the visual and motor bursts are clearly dissociated.

Minor comments

1. In Fig.1A, the label of horizontal axis "Direction from horizontal of movement..." seems odd. Please check this.
2. The insets in the first row of Fig.4 needs labeling of horizontal and vertical axes. The horizontal axis may be the saccade amplitude as in the others, but the vertical axis should be labeled by "ms".

Reviewer #2 (Remarks to the Author):

This paper studies the way in which activity within the superior colliculus (SC) relates (or not) to the kinematics of saccadic eye movements. By comparing saccade-related burst for saccades directed obliquely into the upper vs lower hemifield, Hafed describes how such activity is considerably higher for downward-directed saccades, despite similar peak velocities. The claim is that such a discrepancy runs counter to recent models for how SC activity influences saccade kinematics, leading to questions about such models and about the functional significance of the magnitude of saccade-related SC activity more generally.

There are positives to the manuscript. The paper is very well written and the figures excellent in quality. Often times on my first read, I found questions that I had in one paragraph were answered in

a subsequent paragraph. Further, the data itself was collected with considerable skill, and is of excellent quality. That being said, I do have reservations about the core assumption underlying the manuscript. I am also not convinced that the manuscript actually studies what it purports to; the measures of both saccade-related activity and saccade parameters strike me as fairly coarse, and the majority of the results (e.g., 4 of the 5 figures) are concerned with the comparative saccade properties across upper vs lower hemifields. I expand on these two points, and a few others, below,

Getting to the heart of the question, is a comparison across upward vs downward hemispheres a fair test of temporal coding of saccade kinematics in the SC? I am not so sure about this. It is not immediately clear to me why upward vs downward saccades would have to be governed by the same mapping of SC burst magnitude to saccade kinematics; after all, SC saccade-related bursts are higher for small vs. large saccades, despite reaching slower peak velocities (e.g., Fig 5 of Goossens and Van Opstal PLOS Comp Biol 2012; Hagedorn's own work on small amplitude saccades has also shown this, I believe). At least in part, the thinking is that this rostro-caudal variation relates in part to a greater synaptic weighting of caudal SC efferents onto premotor reticular burst neurons. Why couldn't there be a similar difference in a stronger weighting of neurons coding upper-field saccades, so that similar velocity movements are produced for comparative lower burst magnitudes? Further, it is suspected that these premotor weights are not absolutely fixed, but can be rapidly altered in scenarios with saccadic adaptation (e.g. see Quessy, Quinet, Freedman J Neurosci 2010). Having different weightings for upward vs downward directed saccades seems quite a plausible mechanism, despite the author's reasoning in the Discussion (lines 469-483). I want to be clear that I think there is still value in describing the asymmetry between SC activity for upper- vs. lower-field saccades, but I remain unconvinced that such a comparison is a fair way to test temporal coding in the first place.

Somewhat surprisingly, the measures described in the manuscript are relatively coarse, for a paper that purports to measure the SC "temporal rate code against instantaneous movement evolution". For example, the temporal aspects of SC activity are quantified only by computing the average firing rate in the 50 ms leading up to saccade onset, and the primary measure for saccade kinematics is peak velocity. Contemporary papers in the field measure both SC activity and movement kinematics in more precise and refined ways (e.g., at the trial-by-trial level throughout the trajectory in Smalianchuk et al 2018; more expansive measures of saccade properties in the work by Goossens and Van Opstal). In such studies, saccade related SC activity is also studied in a tighter window relative to saccade onset and offset, and one that changes with saccade duration. While the observation of higher saccade-related activity for downward saccades will persist regardless of how things are quantified, I recommend the author do more to justify the choice of the measures of SC activity (e.g., why 50 ms? Why not ~20 ms, given thoughts on the minimum efferent lag between SC activity and the movement periphery?) and saccade properties. I also recommend backing off claims about "instantaneous movement evolution" unless things are really looked at on a ms-by-ms basis. Relatedly, the author should describe the convolution function for how spikes were converted to a spike density function (apologies if I missed this in the Methods).

As mentioned above, 4 of the 5 figures in the manuscript don't actually show SC activity. The only depiction of neural activity is in Figure 1, and even this only shows activity averaged across many neurons. Given the claims in the manuscript, I think it is fair to expect a more fine-grained examination of SC activity, at least at a neuron-by-neuron if not trial-by-trial level.

In the final point of the Discussion, the author finishes with somewhat vague statements about "other functional roles" that SC movement-related activity could be fulfilling, referencing recent important work about links with high-level perceptual and cognitive behaviours. I think more could be done here to try to establish such links; is there any evidence for example that perceptual and cognitive behaviours vary across upper- vs lower hemifields? Another potential avenue that could be considered are other functional differences between how vision from the upper vs lower hemifields are used in real life, with peripersonal explorations and perhaps eye-hand coordination being more common in the lower visual field.

**Responses to Reviewer Comments on:
“Superior colliculus saccade motor bursts do not dictate movement
kinematics”
COMMSBIO-21-2496
August 17, 2022**

We thank both reviewers very much for carefully evaluating our study. We have now taken all comments into account when revising the manuscript, and **we have also added extensive new neurophysiological experiments and analyses**. We believe that our added experiments **directly address all** comments, and they also support our original interpretations.

Please note that the additional extensive experiments resulted in the addition of **new authors** to the original manuscript.

All edits in the revised manuscript are highlighted with a different font color.

Executive Summary of the Revisions and Responses to Reviewer Comments

We have now introduced **major revisions**, including the addition of new recording data from **>110 new neurons in 3 monkeys, two of which being new monkeys whose data were not part of the original manuscript’s SC motor burst measurements**. Our revised manuscript, thus, now has **neural and behavioral data from a total of 4 monkeys**, all of which demonstrating **consistent results** and **across different behavioral contexts**.

In our new experiments, we have also **directly compared vector-matched saccades under visual and memory-guided conditions**, exactly as suggested. The results directly supported our original interpretations.

We have also modified the text **throughout the manuscript**, to clarify our message in the best possible way.

Please also note that the current literature counts spikes during saccades as the measure of the rate code (e.g. several classic studies of Van Opstal and colleagues). This is essentially exactly what we did in our study (as we clarify in more detail below and in the manuscript), so our analyses **do** replicate the standard analyses in the field.

Concerning the title, note that Reviewer 1 specifically suggested **comparing vector-matched saccades to justify the title**. Because we have now performed these experiments exactly, and because we have replicated the standard neural analyses in the field and got results inconsistent with the idea of kinematic control of saccades by the SC motor bursts, we have decided not to change our title. **The title is descriptive of our results**. We also have other work from different scientific contexts strongly supporting our interpretations in the current study (e.g. Baumann et al., COSYNE, 2022; Baumann et al., VSS, 2022; Hafed et al., NCM, 2022). There is also plenty of other work in the literature consistent with our interpretation (e.g. Edelman &

Goldberg, 2001 to Buonocore et al., 2021). We, therefore, prefer to keep our title unchanged. We believe that our work can motivate new experiments to explore the topic further, and this is a good thing for science. For example, the great debate in the literature about “intention” versus “attention” in the parietal cortex in the 1990’s and early 2000’s was tremendously informative for students of the field at the time, and having varying opinions (and varying titles) is not at all a bad thing for advancing science forward. We would be very happy if our title (and study) were to motivate new exciting experiments on the topic.

Finally, and as stated above, the extensive revisions that we have introduced required revising the author list; these revisions also explain why we took slightly longer than usual to finish our revision.

Responses to Reviewer 1

In this article by Ziad Hafed, the author systematically compared the saccade-related activity of the superior colliculus (SC) neurons, and kinematics of eye movements between the saccades directed to the upward and downward directions. It was found that the presaccadic bursts were stronger for downward saccades (interestingly, it was opposite to the visual responses in his previous study (Hafed and Chen 2016)), even though the saccade kinematics was very similar. Based on these findings, the author concluded that the SC motor bursts do not dictate the movement kinematics as indicated in the title of this paper.

Similar conclusion has been suggested in previous studies on metrically similar executed saccades. In this study, the authors made comparison between the two different saccade vectors and obtained the evidence which complements the proposals made by preceding studies. The analysis is based on a big database of the neuronal and behavioral data and the results are convincing.

OUR RESPONSE: Thank you very much for reviewing our manuscript.

We appreciate your comments, which have motivated us to add **extensive new experiments with vector-matched saccades but different movement kinematics**. Please see our responses below for details; we also include pointers to the new results in the actual revised manuscript.

Having said so, I have some problems in the conclusions of this article, one on conceptual aspect, and the other on the data presentation.

First, as I first read the title of this paper, I imagined that this study is on the comparison between saccade related activity and saccade kinematics for the same saccades as in preceding other studies. And as I read the content of this article, I felt somewhat awkward because the direct conclusion drawn from this article should be that the input-output gain of the upward and downward premotor saccade generator circuits downstream of the SC (presumably in the meso-diencephalic junction?) is different. Interestingly, this would raise another interesting question that the gain of horizontal saccade generator should be different depending on whether the saccade has the upward or downward component (this is just a comment). I think that the present paper supports the concept that the saccade kinematics are not solely

determined by the SC, but primarily by the premotor circuits. Therefore, I think the present title is a bit too strong and general, for the conclusion directly drawn from this article by itself.

OUR RESPONSE: We have now added new experiments in which we recorded from >110 neurons from 3 different monkeys (2 being completely new to the recording component of the whole study and 1 providing an additional recording dataset to the one that was in the original version). In these new experiments, we explicitly focused on **vector-matched saccades towards response field hotspots**, and we had a **behavioral manipulation that systematically altered movement kinematics**. We found no correlation between the kinematic effect of our manipulation and the neural SC burst modulations. In fact, approximately one quarter of all neurons **increased** their saccade-related activity when **saccade speed was reduced**. Moreover, when we recorded multiple neurons simultaneously, we found a diversity of burst modulations accompanying a single kinematic alteration. These results are described in the responses to the next comment below, and also in the **new Figs. 6-8 and lines 434-606 (plus their associated Introduction, Discussion, and Methods text)**.

Because of the new experiments, and in line with lots of other evidence that we now have from other scientific contexts (e.g. Baumann et al., COSYNE, 2022 and Baumann et al., VSS, 2022; Haged et al., NCM, 2022), we believe that our title is suitable and descriptive of our results. We really hope that this title (and our study) will motivate future exciting experiments on the topic.

Please also note that we have additionally modified our text to clarify our original reason for comparing upper and lower visual field saccades (**e.g. please see lines 31, 63-64, 83-85, 106-111, 234-238, 437-444, plus the Discussion**). Briefly, models of saccade kinematic control by SC motor bursts are **agnostic** of saccade direction and have only related movement amplitudes to burst strengths. Therefore, demonstrating different bursts with similar kinematics for **amplitude-matched saccades of different directions** was important in our opinion; it provided an important test of the kinematic control hypothesis (please also see our responses to Reviewer 2 below on this matter).

Second, the author described that the kinematics of delayed and memory-guided saccades are also the same between the upward and downward saccades. But if the author emphasizes that, he should show the neuronal firing data also for them. This is particularly important, because the Fig.1 is on the step visually guided saccades, where the visual and motor bursts are not completely dissociated. Therefore, to draw clear conclusion on the “motor bursts”, the authors should show the data on the delayed or memory-guided saccades where the visual and motor bursts are clearly dissociated.

OUR RESPONSE: We apologize for any confusion that may have existed in the original manuscript. However, the **Fig. 1 data were actually from the delayed visually-guided saccade task**. We did **not** use step visually guided saccades in any neurophysiological results in our manuscript, exactly because the visual and motor bursts become too close to each other in time, as you have correctly stated. When we used a step paradigm for visually guided saccades, it was in the behavioral experiments, in order to allow measuring saccadic reaction times in dataset 2. We

have now clarified this more in the revised manuscript: **e.g. please see lines 169-172, 238, 350-351, 806-809.**

Having said that, your comment opened our eyes to the tremendous value in **comparing delayed visually-guided and memory-guided saccades using matched movement vectors**. Therefore, in new experiments in 3 monkeys, we recorded SC motor bursts for **vector-matched saccades** of the two types. Behaviorally, the movement vectors were matched and towards the response field hotspot, but the movement kinematics were very systematically different – memory-guided saccades were systematically slower than their visually-guided counterparts. Neuronally, for a subset of our new experiments, we used electrode arrays, which allowed us to sometimes capture multiple neurons simultaneously during the same trials. We found a diversity of SC movement burst strength changes as a result of the behavioral manipulation (e.g. the **new Figs. 6 and 7 of the revised manuscript**), even though the kinematic effect was of a single direction (reduction of speed for memory-guided saccades). In fact, with memory-guided saccades, peak velocity was systematically reduced relative to delayed, visually-guided saccades, but approximately half of the neurons had burst strength modulations violating this tendency; moreover, almost one quarter of the neurons actually increased their burst strengths for memory-guided saccades. This meant that across the population, there was **no correlation** between saccade speed effects and SC burst strength effects (**please see Figs. 6-8 of the revised manuscript and their associated text; e.g. lines 434-606, plus their associated Introduction, Discussion, and Methods text**). We view this as strong evidence supporting our interpretation (and title). As also mentioned above, we have additional upcoming manuscripts covering different scientific topics, but also showing a clear dissociation between SC saccade-related burst strength and movement kinematics (e.g. Baumann et al., COSYNE, 2022; Baumann et al., VSS, 2022; Hafeed et al., NCM, 2022); this is in addition to other published evidence by us and others (e.g. Edelman & Goldberg, 2001; Peel et al., 2020; Buonocore et al., 2021), as we mentioned in our Discussion section.

In addition to Figs. 6-8, please also see the following lines in the revised manuscript describing our new experiments and results: **lines 434-606, 616-618, 627-629, 671-673, 699-701, 703-704, 868-908, 975-1041.**

Minor comments

1. In Fig.1A, the label of horizontal axis “Direction from horizontal of movement...” seems odd. Please check this.

OUR RESPONSE: We have now chosen better wording. Please see the new revised Fig. 1a.

2. The insets in the first row of Fig.4 needs labeling of horizontal and vertical axes. The horizontal axis may be the saccade amplitude as in the others, but the vertical axis should be labeled by “ms”.

OUR RESPONSE: Thank you. We agree that this is important. We have now fixed the figure.

Responses to Reviewer 2

This paper studies the way in which activity within the superior colliculus (SC) relates (or not) to the kinematics of saccadic eye movements. By comparing saccade-related burst for saccades directed obliquely into the upper vs lower hemifield, Hafed describes how such activity is considerably higher for downward-directed saccades, despite similar peak velocities. The claim is that such a discrepancy runs counter to recent models for how SC activity influences saccade kinematics, leading to questions about such models and about the functional significance of the magnitude of saccade-related SC activity more generally.

There are positives to the manuscript. The paper is very well written and the figures excellent in quality. Often times on my first read, I found questions that I had in one paragraph were answered in a subsequent paragraph. Further, the data itself was collected with considerable skill, and is of excellent quality. That being said, I do have reservations about the core assumption underlying the manuscript. I am also not convinced that the manuscript actually studies what it purports to; the measures of both saccade-related activity and saccade parameters strike me as fairly coarse, and the majority of the results (e.g., 4 of the 5 figures) are concerned with the comparative saccade properties across upper vs lower hemifields. I expand on these two points, and a few others, below,

OUR RESPONSE: Thank you very much for reviewing our manuscript. We have made every effort to genuinely address all reviewer comments. This included the addition of **extensive new recording experiments from 3 monkeys and >110 neurons, comparing vector-matched saccades having clearly different kinematics**. We have also clarified our text descriptions throughout the manuscript. We are strongly convinced that our revised manuscript deserves to be part of the literature on SC motor bursts, and we really hope that you will agree with our assessment. We also hope that our work will motivate exciting new future research on the topic.

Getting to the heart of the question, is a comparison across upward vs downward hemispheres a fair test of temporal coding of saccade kinematics in the SC? I am not so sure about this. It is not immediately clear to me why upward vs downward saccades would have to be governed by the same mapping of SC burst magnitude to saccade kinematics; after all, SC saccade-related burst are higher for small vs. large saccades, despite reaching slower peak velocities (e.g., Fig 5 of Goossens and Van Opstal PLOS Comp Biol 2012; Hafed's own work on small amplitude saccades has also shown this, I believe). At least in part, the thinking is that this rostro-caudal variation relates in part to a greater synaptic weighting of caudal SC efferents onto premotor reticular burst neurons. Why couldn't there be a similar difference in a stronger weighting of neurons coding upper-field saccades, so that similar velocity movements are produced for comparative lower burst magnitudes? Further, it is suspected that these premotor weights are not absolutely fixed, but can be rapidly altered in scenarios with saccadic adaptation (e.g. see Quessy, Quinet, Freedman J Neurosci 2010). Having different weightings for upward vs downward directed saccades seems quite a plausible mechanism, despite the author's reasoning in the Discussion (lines 469-483). I want to be clear that I think there is still value in

describing the asymmetry between SC activity for upper- vs. lower-field saccades, but I remain unconvinced that such a comparison is a fair way to test temporal coding in the first place.

OUR RESPONSE: Thank you for this important comment. We first respond to it here briefly, and we then describe the actual measures taken in the revised manuscript to address it.

The general assumption in the literature is that there are no directional anisotropies in the SC map for saccade control. Indeed, the classic work of many colleagues, like Van Opstal and others, focuses only on one saccade direction (e.g. horizontal); very often, this is achieved by collapsing across different saccade directions from the actual experiments and analyzing the amplitude effects. Thus, such collapsing across saccade directions reflects the general assumption that it is only amplitude that matters for the mapping that takes place from the SC to the brainstem. Therefore, if we take the same logic that is now existent in the literature, then we have a **well-motivated testable hypothesis** about SC motor bursts and saccade kinematics: matching the amplitudes of saccades for upper and lower visual fields should yield the same saccade bursts. This is not what we found. In other words, we generated a hypothesis based on the accepted norms of the field that saccade direction should not matter, and we showed that it indeed matters. This was, in our opinion, a standard scientific approach, and it also helped us to document an asymmetry in SC motor bursts, which we agree with you is very interesting to document.

Having said that, we are now convinced that **comparing vector-matched saccades to response field hotspots is indeed an important additional test of our hypothesis**. Therefore, we have now collected new data from **3 monkeys directly comparing visual and memory-guided saccades**. We ensured that the vectors were matched across the two types of saccades (please see the **new Figs. 6-8**), and we found that the kinematics were different (as might be expected); memory-guided saccades were systematically slower than visually-guided saccades. We then checked individual neurons and asked whether the alterations in motor burst strengths were all in the same direction as the kinematic alterations. There was no systematic reduction of motor bursts in SC neurons with the slower memory-guided saccades. In fact, almost one quarter of the neurons **increased** their motor burst strengths for the slower memory-guided saccades when compared to the faster visually-guided saccades. And, when we recorded multiple neurons simultaneously in the same sessions, there was a clear diversity of motor bursts for the very same trials; some neurons had stronger motor bursts for visually-guided saccades, some neurons had stronger motor bursts for memory-guided saccades, and some neurons had similar motor bursts. Overall, there was no correlation between the behavioral kinematic effect and the neural effect (please see the **new Fig. 8**).

Therefore, both using an existing assumption in the literature that saccade direction does not matter as well as using vector-matched saccades but with strongly different kinematics, we found results that are inconsistent with the idea that SC motor bursts dictate saccade kinematics. This is in addition to other evidence in the literature that we have alluded to in our manuscript and above.

The revised manuscript describes the above points as follows:

- Clarification of our motivation for comparing upper and lower visual field saccades: **lines 82-85, 106-112, 234-238, 437-443, 614**
- The new vector-matched saccade recordings: **Figs. 6-8 and lines 434-606 (plus the associated Introduction, Discussion, and Methods text)**

Somewhat surprisingly, the measures described in the manuscript are relatively coarse, for a paper that purports to measure the SC “temporal rate code against instantaneous movement evolution”. For example, the temporal aspects of SC activity are quantified only by computing the average firing rate in the 50 ms leading up to saccade onset, and the primary measure for saccade kinematics is peak velocity. Contemporary papers in the field measure both SC activity and movement kinematics in more precise and refined ways (e.g., at the trial-by-trial level throughout the trajectory in Smalianchuk et al 2018; more expansive measures of saccade properties in the work by Goossens and Van Opstal). In such studies, saccade related SC activity is also studied in a tighter window relative to saccade onset and offset, and one that changes with saccade duration. While the observation of higher saccade-related activity for downward saccades will persist regardless of how things are

quantified, I recommend the author do more to justify the choice of the measures of SC activity (e.g., why 50 ms? Why not ~20 ms, given thoughts on the minimum efferent lag between SC activity and the movement periphery?) and saccade properties. I also recommend backing off claims about “instantaneous movement evolution” unless things are really looked at on a ms-by-ms basis. Relatedly, the author should describe the convolution function for how spikes were converted to a spike density function (apologies if I missed this in the Methods).

OUR RESPONSE: We have now been much more careful in the text with respect to terms like place code and rate code, and so on. For example, we have now elected to always use “rate code” in the revised manuscript instead of more nuanced, wording as you suggested (please see, for example, **lines 58, 61, 65, 66, 74, and so on**; all uses of the “rate code” are now highlighted in blue in the revised manuscript).

We have also clarified the convolution kernel used for firing rate estimates (**please see lines 913-914**).

Please also note that we used essentially the exact same analysis method as Van Opstal’s work. There, they count spikes, and our averages or peaks of firing rates are essentially counting spikes within an interval. We have now clarified this (**e.g. please see lines 968-972**).

As you also stated, our argument is that if the original hypothesis is invalidated with the simplest method possible (spike counts), then this is a very powerful result. That is why we felt that our approach is the best one here. Once again, **please see lines 968-972**.

Having said that, we have now added extensive new experiments with vector-matched visually-guided and memory-guided saccades (**Figs. 6-8 and lines 434-606; plus the associated Introduction, Discussion, and Methods text**). We believe that the results from these experiments are compelling, and they agree with a lot of evidence from the literature that we have alluded to above and in the

manuscript. We also know that our own upcoming papers will show even more evidence arguing that movement kinematics are dissociated from SC motor bursts (as mentioned above).

Please also note that in the new experiments, we used a peri-saccadic interval as suggested (not exclusively a pre-movement interval), and we reached similar conclusions to our original ones. For example, **please see lines 969-972**.

Finally, regarding Smalianchuk et al., we are concerned about correlating a velocity curve with a firing rate density curve. Such curves are rendered very similar to each other due to heavy averaging (kernel convolution in the case of firing rate densities, and temporal smoothing in the case of eye velocity estimates). It is, therefore, a bit risky (in our opinion) to correlate these curves, and we would much rather stick to measures related to spike counts like we (and others) did. And, as you said, the differences that we saw between the upper and lower visual fields would still be present. Similarly, the results of our new experiments with vector-matched visual and memory-guided saccades would persist with this approach. Finally, we think that the trial-to-trial variability that Smalianchuk et al. saw could be related to other factors, like the trial-to-trial variability in target visibility or animal motivation.

To summarize, the combination of our new experiments, text edits, and similarity of our analysis techniques to standard methods in the field make us confident in our interpretation.

As mentioned above, 4 of the 5 figures in the manuscript don't actually show SC activity. The only depiction of neural activity is in Figure 1, and even this only shows activity averaged across many neurons. Given the claims in the manuscript, I think it is fair to expect a more fine-grained examination of SC activity, at least at a neuron-by-neuron if not trial-by-trial level.

OUR RESPONSE: We have now followed this advice directly, and we have collected extensive **new data from 3 monkeys and >110 neurons**. In these new experiments, we have explicitly compared motor bursts in visually-guided and memory-guided saccades having different kinematics, and we have done neuron-by-neuron analyses as suggested. **Please see Figs. 6-8 and lines 434-606; plus the associated Introduction, Discussion, and Methods text.**

In the final point of the Discussion, the author finishes with somewhat vague statements about "other functional roles" that SC movement-related activity could be fulfilling, referencing recent important work about links with high-level perceptual and cognitive behaviours. I think more could be done here to try to establish such links; is there any evidence for example that perceptual and cognitive behaviours vary across upper- vs lower hemifields? Another potential avenue that could be considered are other functional differences between how vision from the upper vs lower hemifields are used in real life, with peripersonal explorations and perhaps eye-hand coordination being more common in the lower visual field.

OUR RESPONSE: We agree, and we have now updated this paragraph as suggested. **Please see lines 717-729.**

Reviewers' comments:

Reviewer #1 (Remarks to the Author):

In response to my comments to the original version of the manuscript, the authors added a new recording data from >110 new neurons in 3 monkeys to compare the SC motor bursts for saccades with the same kinematics. I admire the great efforts made by the authors and am convinced by the conclusion. Reading this new version of the manuscript, I wish to raise a few minor issues, which would improve the clarity of the article to a wide range of readers.

1. The analysis in Fig.1 covers the neurons with a wide range of eccentricities and direction of "hotspots". This analysis is validated by our knowledge that the peak firing rate of the SC motor bursts are similar for saccades toward their hotspots. This knowledge is shared by the scientists working on the SC but not always to a wider range of audience (for instance, the primary motor cortex does not have this way of coding). I understand that the authors narrowed down the hotspots to 5-15 deg eccentricities and 0 ± 30 deg directions in Fig. 1C, but still so. I recommend the authors to make this point clear when they explain the results of this figure.

2. In Discussion, the authors mentioned the possibility "structures downstream from the SC would compensate for the SC asymmetry" as "counter argument" (line 668-676), but it is hard for me to understand this. I think it was "complementary argument". What I meant in my comments to the original version was that the saccade motor commands are eventually in the rate code in the brainstem saccade generator circuits including motoneurons. Therefore, asymmetry in the SC output neuron activity should be transformed to the rate code for the saccade kinematics somewhere between the SC and brainstem saccade generators. Is my understanding wrong?

Reviewer #2 (Remarks to the Author):

I enjoyed reading the revised version of this manuscript, and thank the authors for the efforts that they have put into it. As is often the case with an interesting dataset, the results raise some intriguing questions for future studies.

The presented data shows that no one mapping relates SC activity to saccade kinematics across all tasks/situations. The first part of the results shows this by comparing mirror-matched oblique saccades to the upper vs lower hemifield – compared to downward saccades, the SC bursts for downwardly oriented saccades are stronger robust even though peak saccade velocity is similar.

In the second, and essentially new, half of the results, the authors carefully vector-match saccades made under different behavioral conditions. As a whole, SC activity and saccade velocity decreased for memory-guided saccades, replicating previous results. The authors add to this by showing that while this finding holds at the population level, a substantial proportion of SC neurons display increased saccade-related activity during slower memory-guided saccades. Again, it's the implications of these results that are the most interesting; left for future studies is the way in which population-levels of activity, either within the SC or perhaps distributed across other structures, are shaped by different behavioral tasks and ultimately integrated by downstream structures. As an aside, the anecdotal examples displaying increased activity during slower saccades tended to have substantially lower peak firing rates to begin with; did this hold true across the sample, and/or (for those neurons recorded simultaneously) was there any tendency for such neurons to be located higher- or lower- in the SC compared to neurons displaying decreased saccade-related activity for slower saccades? Such analyses may start to reveal functional or anatomical differences between SC neurons displaying saccade-related activity.

**Responses to Reviewer Comments on:
“Superior colliculus saccade motor bursts do not dictate movement
kinematics”
COMMSBIO-21-2496
October 13, 2022**

We thank the reviewers once again for their continued support. We have now completed the few remaining minor edits to the manuscript, as detailed below. We trust that the manuscript is now ready for publication.

Responses to Reviewer 1

In response to my comments to the original version of the manuscript, the authors added a new recording data from >110 new neurons in 3 monkeys to compare the SC motor bursts for saccades with the same kinematics. I admire the great efforts made by the authors and am convinced by the conclusion. Reading this new version of the manuscript, I wish to raise a few minor issues, which would improve the clarity of the article to a wide range of readers.

1. The analysis in Fig.1 covers the neurons with a wide range of eccentricities and direction of “hotspots”. This analysis is validated by our knowledge that the peak firing rate of the SC motor bursts are similar for saccades toward their hotspots. This knowledge is shared by the scientists working on the SC but not always to a wider range of audience (for instance, the primary motor cortex does not have this way of coding). I understand that the authors narrowed down the hotspots to 5-15 deg eccentricities and 0-±30 deg directions in Fig. 1C, but still so. I recommend the authors to make this point clear when they explain the results of this figure.

OUR RESPONSE: Thank you very much for helping us to clarify the remaining points. We appreciate such help very much.

Concerning this first comment, yes, we have now followed your advice and edited the text appropriately. We did so near the beginning of the Results section. Please see lines 136-137, 169-172, 181-182. Please also note that we matched eccentricities in our comparison between upper and lower visual field neurons, which was the critical comparison of interest for us.

2. In Discussion, the authors mentioned the possibility “structures downstream from the SC would compensate for the SC asymmetry” as “counter argument” (line 668-676), but it is hard for me to understand this. I think it was “complementary argument”. What I meant in my comments to the original version was that the saccade motor commands are eventually in the rate code in the brainstem saccade generator circuits including motoneurons. Therefore, asymmetry in the SC output neuron activity should be transformed to the rate code for the saccade kinematics somewhere between the SC and brainstem saccade generators. Is my understanding wrong?

OUR RESPONSE: We fully agree, and we do apologize for not articulating this idea more clearly in the text of the previous revision. We have now rewritten the whole paragraph as suggested (please see lines 518-535).

Responses to Reviewer 2

I enjoyed reading the revised version of this manuscript, and thank the authors for the efforts that they have put into it. As is often the case with an interesting dataset, the results raise some intriguing questions for future studies.

The presented data shows that no one mapping relates SC activity to saccade kinematics across all tasks/situations. The first part of the results shows this by comparing mirror-matched oblique saccades to the upper vs lower hemifield – compared to downward saccades, the SC bursts for downwardly oriented saccades are stronger robust even though peak saccade velocity is similar.

In the second, and essentially new, half of the results, the authors carefully vector-match saccades made under different behavioral conditions. As a whole, SC activity and saccade velocity decreased for memory-guided saccades, replicating previous results. The authors add to this by showing that while this finding holds at the population level, a substantial proportion of SC neurons display increased saccade-related activity during slower memory-guided saccades. Again, it's the implications of these results that are the most interesting; left for future studies is the way in which population-levels of activity, either within the SC or perhaps distributed across other structures, are shaped by different behavioral tasks and ultimately integrated by downstream structures. As an aside, the anecdotal examples displaying increased activity during slower saccades tended to have substantially lower peak firing rates to begin with; did this hold true across the sample, and/or (for those neurons recorded simultaneously) was there any tendency for such neurons to be located higher- or lower- in the SC compared to neurons displaying decreased saccade-related activity for slower saccades? Such analyses may start to reveal functional or anatomical differences between SC neurons displaying saccade-related activity.

OUR RESPONSE: Thank you very much. We are indeed very intrigued about the implications of these results for future studies, and we hope to publish highly interesting follow up observations very soon.

Concerning the final aside: the neurons increasing their activity for the slower saccades did indeed tend to have slightly lower firing rates than the other neurons, but we observed them at all depths in our experiments. We have now mentioned this briefly in lines 456-458 of the revised manuscript.